# Heavy-Tailed Diffusion with Denoising Lévy Probabilistic Models

**Dario Shariatian[1], Umut Simsekli[1], Alain Durmus[2]**
[1]INRIA - Department of Computer Science, PSL Research University, Paris, France
[2]École Polytechnique - CMAP, IP Paris, Palaiseau, France
`{dario.shariatian, umut.simsekli}@inria.fr`
`alain.durmus@polytechnique.edu`

## Abstract

Exploring noise distributions beyond Gaussian in diffusion models remains an open challenge. While Gaussian-based models succeed within a unified SDE framework, recent studies suggest that heavy-tailed noise distributions, like $\alpha$-stable distributions, may better handle mode collapse and effectively manage datasets exhibiting class imbalance, heavy tails, or prominent outliers. Recently, Yoon et al. (NeurIPS 2023), presented the Lévy-Itô model (LIM), directly extending the SDE-based framework to a class of heavy-tailed SDEs, where the injected noise followed an $\alpha$-stable distribution, a rich class of heavy-tailed distributions. However, the LIM framework relies on highly involved mathematical techniques with limited flexibility, potentially hindering broader adoption and further development. In this study, instead of starting from the SDE formulation, we extend the denoising diffusion probabilistic model (DDPM) by replacing the Gaussian noise with $\alpha$-stable noise. By using only elementary proof techniques, the proposed approach, *Denoising Lévy Probabilistic Model* (DLPM), boils down to vanilla DDPM with minor modifications. As opposed to the Gaussian case, DLPM and LIM yield different training algorithms and different backward processes, leading to distinct sampling algorithms. These fundamental differences translate favorably for DLPM as compared to LIM: our experiments show improvements in coverage of data distribution tails, better robustness to unbalanced datasets, and improved computation times requiring smaller number of backward steps.

## 1 Introduction

The evolution of generative models has introduced several approaches, with diffusion models emerging as one of the most prominent. These models transform a data distribution into a Gaussian distribution via a forward noising process and then learn to reverse it. The foundational work on denoising in this context was presented by Sohl-Dickstein et al. (2015), where the goal is to reverse a Markov chain that progressively adds Gaussian noise to the data. This framework culminated in denoising diffusion probabilistic models (DDPM) by Ho et al. (2020), which demonstrated state-of-the-art performance in image generation, while drawing connections to score matching techniques (Song & Ermon (2020)). A unified theoretical framework, based on stochastic differential equations (SDEs), further integrated score matching and the denoising framework (Song et al. (2021)). Various generative models build up on this framework, improving its performance (Dhariwal & Nichol (2021a); Karras et al. (2022)).

Despite their success, diffusion models exhibit limitations, such as requiring a large number of steps (Ho et al. (2020)) and empirically struggling with imbalanced datasets (Zhang et al. (2024); Yoon et al. (2023)), often leading to mode collapse (Dhariwal & Nichol (2021a); Deasy et al. (2022)). Tackling heavy-tailed datasets also presents a challenge for diffusion models, as the finite-time diffusion processes generate finite variance data distributions, which are ill-suited for modeling heavy-tailed data. Such datasets, like financial datasets, could benefit from extensions beyond the Gaussian setting, as demonstrated in Borak et al. (2005). Moreover, although many techniques exist to improve quality at the expense of diversity (Dhariwal & Nichol (2021b); Song et al. (2023)), by only

using Gaussian noise, the methods typically require highly nontrivial setups in order to obtain improved diversity while maintaining a reasonable quality (e.g., FID score) (Sadat et al. (2024); Nobis et al. (2024)).

Several approaches have been explored to address the limitations of diffusion models, particularly through the use of non-Gaussian, heavy-tailed noise distributions. The motivation behind this is that heavy-tailed distributions, which can take on larger values, may reduce the number of sampling steps and better capture multimodal data distributions by identifying isolated modes through large noise injections (Yoon et al. (2023)). These distributions are also capable of modeling extreme events or rare occurrences in the tails, making them suitable for tasks such as audio generation (Chen et al. (2020); Kong et al. (2021)), where rare but important variations in amplitude or pitch (e.g., prosodies) can enhance sample quality and diversity. An early attempt at using heavy-tailed noise distributions was made by Nachmani et al. (2021), who replaced Gaussian noise with Gamma-distributed noise, reducing the number of diffusion steps and improving data diversity. Similarly, Deasy et al. (2022) employed generalized Gaussian distributions in a score-matching formulation to improve robustness on unbalanced datasets. However, despite these promising directions, the performance boosts are limited, since both approaches fail to provide a time-reversal formula, either using annealed Langevin dynamics when the score is available, or using heuristics based on the Gaussian approach.

Very recently, partially inspired by Simsekli (2017); Huang et al. (2020b), who employ heavy-tailed SDEs in Monte Carlo sampling for challenging distributions, Yoon et al. (2023) extended the SDE framework by replacing the light-tailed Brownian motion with a heavy-tailed driving process, introducing the Lévy-Itô model (LIM), improving performance on image data, particularly for unbalanced datasets, offering gains in metrics like FID and diversity. The tail index $\alpha$ controls the degree of tail heaviness, enabling tunable performance based on the characteristics of the data. When $\alpha = 2$ the noising process reduces to the Brownian motion (light-tailed), however, whenever $\alpha < 2$, the process becomes heavy-tailed with infinite variance (notably, heavier-tailed than the distributions explored by Nachmani et al. (2021), Deasy et al. (2022)).

While injecting heavy-tailed, infinite-variance noise might seem natural when the data distribution itself is also heavy-tailed, Yoon et al. (2023) further illustrated that the heavy-tailed noise can also be beneficial when sampling from compactly supported data distributions (e.g., generating images), especially in the presence of class imbalances (i.e., the large 'jumps' introduced by the heavy tails can help finding weakly represented modes). Indeed, perhaps being counter-intuitive, it has been shown that a heavy-tailed process can indeed converge to a light-tailed distribution with appropriate care Simsekli (2017); Simsekli et al. (2020); Huang et al. (2020a), making them suitable for sampling from a broad range of data distributions.

**Motivation.** While LIM demonstrates promising results, the technical complexity of the time-reversed SDE presents significant challenges. The Lévy process with $\alpha < 2$ has discontinuous paths and no variance, preventing the use of standard analysis tools. The proof techniques rely on fractional calculus and estimations for pseudo-differential operators, which might not be accessible for the broader community of diffusion-based generative models. While the theory is elegant, we argue that the highly technical nature of LIM, originated due to the use of continuous-time processes, might hinder its development. For instance, it is highly non-trivial to use arbitrary noise schedules.

Moreover, the loss function used in LIM presents some shortcomings: the theory requires a squared $\ell_2$ loss, assuming the loss remains finite for the considered neural network. However, this may not always hold, as the noise term is heavy-tailed and admits no variance, potentially leading to infinite loss values. As a result, LIM experiments must revert to an $\ell_1$ loss for stable training, suggesting that the original loss function may indeed be unworkable. Additionally, LIM is constrained to isotropic noise, further limiting its flexibility.

To overcome these issues, we propose a simpler yet effective alternative for incorporating heavy-tailed distributions, focusing on a discrete-time framework that leverages more elementary mathematical tools while maintaining performance improvements over Gaussian-based models.

**Contributions.** As opposed to LIM which extended the SDE-based framework, here, we take a step back and directly work on the discrete-time DDPM process and replace the Gaussian noise with $\alpha$-stable noise. More precisely, we propose the following Markov process as noising process:

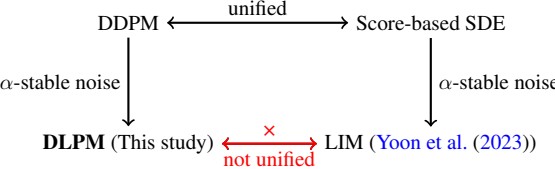

Figure 1: Illustration of available methods.

$$X_t = \gamma_t X_{t-1} + \sigma_t \epsilon_t^{(\alpha)}, \qquad (1)$$

where $\epsilon_t^{(\alpha)}$ follows a multivariate $\alpha$-stable distribution. When $\alpha = 2$, the process recovers the standard Gaussian DDPM, but for $\alpha < 2$, it introduces heavy-tailed noise with infinite variance. A comparison between DLPM and LIM is provided in Figure 1. Our contributions are as follows.

• **Simplified mathematical framework.** Leveraging a property of stable distributions (cf. Theorem 1), we decompose $\epsilon_t^{(\alpha)}$ into a product of a one-dimensional random variable and a Gaussian vector. This transformation reduces the forward process to a Gaussian one with modified scaling, allowing us to approximate the reverse process using only elementary tools. We call the resulting generative model Denoising Lévy Probabilistic Model (DLPM).

• **Extension to deterministic sampling.** Building on the DDIM framework (Song et al. (2020)), we introduce a deterministic sampler for DLPM, termed DLIM, which further reduces the number of sampling iterations, boosting efficiency.

• **Compatibility with existing methods.** DLPM maintains compatibility with existing DDPM implementations, requiring only minor modifications, making it a practical and flexible alternative to LIM. This simplicity extends for instance to noise schedules, which are difficult to manipulate in the continuous-time LIM framework.

• **Distinct Algorithms from LIM.** Unlike the Gaussian case, where DDPM and the score-based SDE formulation are two sides of the same coin (Song et al. (2021)), we show that DLPM and LIM result in distinct training algorithms and backward processes. Importantly, as these heavy-tailed distributions admit no variance, we cannot simply use the square $\ell_2$ loss, problem which DLPM carefully addresses.

• **Improved performances.** Thanks to our conditionally Gaussian representation strategy, our networks are only modeling conditional densities given the heavy-tailed variables. Hence the network is not learning a heavy-tailed distribution directly, but a light-tailed conditional distribution, intuitively an easier task. These differences work in favor of DLPM across several performance aspects, particularly where heavy-tailed noise injections are already known to offer advantages. Our experiments show that DLPM provides (i) better coverage of the tails of the data distribution, (ii) improved generation of unbalanced datasets, and (iii) faster computation times, requiring fewer backward steps.

## 2    BACKGROUND ON $\alpha$-STABLE DISTRIBUTIONS

The family of $\alpha$-stable distributions appears as the limiting distribution in the generalized central limit theorem (Gnedenko & Kolmogorov (1968)). In the one dimensional case, an $\alpha$-stable distributed random variable $X$ is defined through its characteristic function (Samorodnitsky et al. (1996)): for $u \in \mathbb{R}$

$$\mathbb{E}\left[e^{iuX}\right] = \exp\{iu\mu - |\sigma u|^\alpha(1 - i\varphi\beta\,\mathrm{sgn}(u))\}, \quad \text{where } \varphi = \begin{cases} \tan(\pi\alpha/2) & \text{if } \alpha \neq 1 \\ -(2/\pi)\log|\sigma u| & \text{otherwise} \end{cases}.$$

Here, (i) $\mu \in \mathbb{R}$ is the location parameter (ii) $\alpha \in (0, 2]$ is the tail index (iii) $\sigma > 0$ the scale parameter (iv) $\beta \in [-1, 1]$ determines the right- or left-skewness, and sgn is the sign function. We denote the $\alpha$-stable distribution by $\mathcal{S}_{\alpha,\beta}(\mu, \sigma)$.

In the case where $\alpha < 1$ and $\beta = 1$, the support of the distribution becomes the positive real line (i.e., the random variable is positive), hence we call this distribution 'positive stable'. On the other hand, in the case where $\beta = 0$, the distribution $\mathcal{S}_{\alpha,0}(\mu, \sigma)$ is symmetric around $\mu$, and denoted by $\mathcal{S}_\alpha(\mu, \sigma)$. Furthermore, in the case $\alpha = 2$, the distribution reduces to a Gaussian $\mathcal{S}_\alpha(\mu, \sigma) =$

$\mathcal{N}(\mu, 2\sigma^2)$, hence it is light-tailed. However, whenever $\alpha < 2$, $\mathcal{S}_\alpha(\mu, \sigma)$ has heavy tails, *i.e.*, the decay rate of its tail distribution satisfies $\mathbb{P}(|X| > r) \sim r^{-\alpha}$ as $r \to \infty$ (see (Nolan, 2020, Theorem 1.2)). This implies that $\mathbb{E}[|X|^p] < \infty$ if and only if $p < \alpha < 2$.

As opposed to Gaussians, there are multiple ways of extending the $\alpha$-stable distributions to the multivariate setting. In this paper, we will be interested in two major cases: (i) the isotropic (also called rotationally invariant)[1] and (ii) the non-isotropic with independent components. These distributions are also defined through their respective characteristic functions. The random variable $X \in \mathbb{R}^d$ is isotropic $\alpha$-stable if its characteristic function is given by: for all $u \in \mathbb{R}^d$, $\mathbb{E}[\exp(iu^\top X)] = \exp(i\mu^\top u - \sigma^\alpha \|u\|^\alpha)$, where $\mu \in \mathbb{R}^d$ is the location parameter and $\sigma I_d$ plays the role of a covariance matrix[2]. We denote it by $X \sim \mathcal{S}_\alpha^i(\mu, \sigma I_d)$. Similarly, $X$ follows the non-isotropic $\alpha$-stable distribution $\mathcal{S}_\alpha^n(\mu, \sigma I_d)$, if for any $u \in \mathbb{R}^d$, $\mathbb{E}[\exp(iu^\top X)] = \exp(i\mu^\top u - \sigma^\alpha \sum_{i=1}^d |u_i|^\alpha)$. While both of these distributions share similar characteristics, such as having power-law tails with the same exponent, the components of the isotropic case are dependent, which results in a significant difference compared to the non-isotropic case, which has independent coordinates. When $\alpha = 2$ both options coincide with a multivariate Gaussian.

The following property of stable distributions will form the backbone of our algorithm.

**Theorem 1** (See (Samorodnitsky et al., 1996, Equation 2.5.3)). *Let $\alpha < 2$, and let $X \sim \mathcal{S}_\alpha^i(\mu, \sigma I_d)$. Then, $X \overset{d}{=} \mu + \sigma A^{1/2} G$, where $\overset{d}{=}$ denotes equality in distribution, $A \sim \mathcal{S}_{\alpha/2,1}(0, c_A)$ is a one-dimensional positive stable random variable with $c_A := \cos^{2/\alpha}(\pi\alpha/4)$, and $G \sim \mathcal{N}(0, I_d)$.*

This theorem shows that a zero-mean, unit-scale isotropic stable random-vector can be equivalently written as the product of a *one dimensional* positive stable random variable and a standard Gaussian random vector. This fundamental property will have a significant impact in terms of incorporating $\alpha$-stable noise in DDPMs in a simple way as, *conditioned on $A$*, the distribution of $X$ is just a Gaussian. We conclude this section by noting that a similar decomposition for the non-isotropic case: if $X \sim \mathcal{S}_\alpha^n(\mu, \sigma I_d)$, then $X \overset{d}{=} \mu + \sigma \mathbf{A}^{1/2} \odot G$, where $\odot$ is the component-wise multiplication and $\mathbf{A}^{1/2} = \{A_i^{1/2}\}_{i=1}^d \in \mathbb{R}^d$ is a vector with i.i.d. components with $A_i \sim \mathcal{S}_{\alpha/2,1}(0, c_A)$.

## 3 DENOISING LÉVY PROBABILISTIC MODELS

In this section, we develop our algorithm by following a similar route to the development on DDPM: we identify the forward and backward processes associated with (1) and construct a variational approximation for the backward chain for sampling. Here, we focus on isotropic $\alpha$-stable noise; however, adaptation to non-isotropic $\alpha$-stable noise is straightforward, as all our derivations rely on Theorem 1, hence it is omitted.

### 3.1 MARKOVIAN FORWARD PROCESS

Recall that DLPM is based on the following recursion (restatement of (1)):

$$X_0 \sim p_\star, \quad \text{and} \quad X_t = \gamma_t X_{t-1} + \sigma_t \epsilon_t^{(\alpha)}, \tag{2}$$

where $p_\star$ is the data distribution and $\{\epsilon_t^{(\alpha)}\}_{t=1}^T$ are independent and distributed as $\mathcal{S}_\alpha^i(0, I_d)$. Thanks to the 'stability' property of $\alpha$-stable distributions, *i.e.*, the sum of two $\alpha$-stable random variables is still $\alpha$-stable (see Appendix B for details), we can explicitly characterize the conditional distribution of $X_t$ given $X_0$. Setting for any $t \in \{1, \ldots, T\}$, $\gamma_{1 \to t} := \prod_{i=1}^t \gamma_t$, and $\sigma_{1 \to t} := (\sum_{i=1}^t (\gamma_{1 \to t} \sigma_i / \gamma_{1 \to i})^\alpha)^{1/\alpha}$, we show in Proposition 4 that:

$$X_t \overset{d}{=} \gamma_{1 \to t} X_0 + \sigma_{1 \to t} \varepsilon_t,$$

where $\varepsilon_t \sim \mathcal{S}_\alpha^i(0, I_d)$. Similarly to DDPM, the noising schedule parameters $\{(\gamma_t, \sigma_t)\}_{t=1}^T$ and the horizon $T$ are set so that the final distribution of $X_T$ is approximately equal to $\mathcal{S}_\alpha^i(0, \sigma_{1 \to t} I_d)$,

---

[1]The noise distribution used in LIM is the isotropic $\alpha$-stable distribution. Note that our framework allows for different types of $\alpha$-stable distributions by following a single mathematical recipe.

[2]in this isotropic case the components are not independent even though the covariance matrix is diagonal.

choosing either (i) $\gamma_{1\to t} \to 0$ as $t$ increases, or (ii) $\gamma_{1\to t} = 1$ and $\sigma_{1\to t}$ increasing with $t$. Following the terminology used in DDPM[3], we refer to schedule (i) as scale preserving and schedule (ii) as scale exploding.

## 3.2 GENERATIVE PROCESS

Once the forward process is run for large enough time-steps $T$, it is clear that $X_T$ will be approximately stable-distributed. Hence, to go back to the data distribution $p_\star$, we now need to time-revert the forward process so that the reversed process can take some $\alpha$-stable noise and generate a sample from $p_\star$, which is our ultimate goal. More precisely, we aim to find a backward process associated to the Markov chain $\{X_t\}_{t=0}^T$, *i.e.*, a Markov chain $\{\overleftarrow{X}_t\}_{t=0}^T$ such that the two processes $\{\overleftarrow{X}_{T-t}\}_{t=0}^T$ and $\{X_t\}_{t=0}^T$ have the same distributions. Since, by (Nolan, 2010, Theorem 1.9), any non-degenerate $\alpha$-stable distribution has a smooth density with respect to the Lebesgue measure, the Markov chain (2) admits a transition density denoted by $k_{t|t-1}^{(\alpha)}$. In addition, $X_t$ admits a density as well, denoted by $p_t^{(\alpha)}$. Then, it can be easily verified that a Markov process starting from $p_T^{(\alpha)}$ and with transition densities, for $t \in \{0, \ldots, T-1\}$, $\overleftarrow{k}_{t-1|t}^{(\alpha)}(x_{t-1}|x_t) \propto p_{t-1}^{(\alpha)}(x_t) k_{t|t-1}^{(\alpha)}(x_t, x_{t-1})$ for any $x_{t-1}, x_t$, is a backward process associated with $\{X_t\}_{t=0}^T$, where $\propto$ denotes equality up to a normalization constant. As in the case of DDPM, this backward transition densities are unfortunately intractable, hence, we will develop a variational scheme for their approximation.

### 3.2.1 APPROXIMATION OF THE BACKWARD TRANSITION DENSITIES IN DDPM

To ease the introduction of our approach, let us first recall the strategy taken in DDPM, which approximates the backward kernels by relying on a variational approximation for $\overleftarrow{k}_{0:T}^{(\alpha)}(x_{0:T}) := p_T^{(\alpha)}(x_T) \prod_{t=T}^1 \overleftarrow{k}_{t-1|t}^{(\alpha)}(x_{t-1}|x_t)$, where $\alpha = 2$ and $x_{0:T} := (x_0, \ldots, x_T) \in \mathbb{R}^{d \times (T+1)}$. More precisely, the goal is to find the 'closest' distribution to $\overleftarrow{k}_{0:T}^{(\alpha)}$ in a family of distributions $\{\overleftarrow{q}_{0:T}^\theta : \theta \in \Theta\}$, indexed by a parameter $\theta$ taking values in some parameter space $\Theta$ (typically taken as a neural network).

The variational family is assumed to have the same decomposition as $\overleftarrow{k}_{0:T}^{(\alpha)}(x_{0:T})$, thus such that $\overleftarrow{q}_{0:T}^\theta(x_{0:T}) := \overleftarrow{q}_T^\theta(x_T) \prod_{t=T}^1 \overleftarrow{q}_{t-1|t}^\theta(x_{t-1}|x_t)$, where $\overleftarrow{q}_T^\theta$ is chosen to be the density of $\mathcal{N}(0, \sigma_{1\to t}^2 \mathrm{I}_d)$ as an approximation of $p_T^{(\alpha)}$. Then, $\theta$ is obtained by minimizing the following objective function (Ho et al., 2020, Equation 5):

$$\mathscr{L}^{\mathrm{D}}(\theta) := \sum_{t=2}^T \mathscr{L}_{t-1}^{\mathrm{D}}(\theta) \quad \text{with} \quad \mathscr{L}_{t-1}^{\mathrm{D}}(\theta) = \mathbb{E}[\mathrm{KL}(k_{t-1|0,t}^{(\alpha)}(\cdot|X_0, X_t)\|\overleftarrow{q}_{t-1|t}^\theta(\cdot|X_t))] , \quad (3)$$

where KL denotes the Kullback-Leibler divergence and $k_{t-1|0,t}^{(\alpha)}$ denotes the conditional density of $X_{t-1}$ given $X_0$ and $X_t$. As $\alpha = 2$ in this case, $k_{t-1|0,t}^{(\alpha)}$ is Gaussian (Ho et al., 2020, Equation 6,7), motivating the choice of Gaussian densities $\overleftarrow{q}_{t-1|t}^\theta(x_{t-1}|x_t)$ as elements of the variational family at hand, since one obtains a closed-form formula for the KL terms, *i.e.*,

$$\overleftarrow{q}_{t-1|t}^\theta(x_{t-1}|x_t) = \phi_d\left(x_{t-1}|\hat{\mathrm{m}}_{t-1}^\theta(x_t), \hat{\Sigma}_{t-1}^\theta(x_t)\right) , \quad (4)$$

where $(x, \mathrm{m}, \Sigma) \mapsto \phi_d(x|\mathrm{m}, \Sigma)$ is the density of the $d$-dimensional Gaussian distribution with mean $\mathrm{m}$ and covariance matrix $\Sigma$, and $\hat{\mathrm{m}}_{t-1}^\theta, \hat{\Sigma}_{t-1}^\theta$ are functions of $x_t$ parameterized by $\theta$. This approach relies on the fact that $k_{t-1|0,t}^{(\alpha)}$ is analytically tractable. Unfortunately, when $\alpha < 2$, it is not the case anymore. We now expose our methodology to address this limitation.

### 3.2.2 A DATA AUGMENTATION APPROACH

To obtain a tractable objective function for learning a variational approximation of the backward transition densities, we rely on a data augmentation approach, which is a classical MCMC technique

---

[3]In the DDPM literature, these noising schedules are referred to as variance preserving, or variance exploding. We use the term 'scale' instead of the variance here, since the variance does not exist when $\alpha < 2$.

(see Brooks et al. (2011), Chapter 10). Consider the Markov chain $Y_0 \sim p_\star$, and for $t \in \{1, \dots, T\}$,

$$Y_t = \gamma_t Y_{t-1} + \sigma_t A_t^{1/2} G_t , \tag{5}$$

where $\{G_t\}_{t=1}^T$ and $\{A_t\}_{t=1}^T$ are independent random variables, distributed according to $G_t \sim \mathcal{N}(0, \mathrm{I}_d)$, and $A_t \sim \mathcal{S}_{\alpha/2,1}(0, c_A)$ with $c_A = \cos^{2/\alpha}(\pi\alpha/4)$. From Theorem 1, $\{Y_t\}_{t=0}^T$ is a Markov chain that admits the same distribution as $\{X_t\}_{t=0}^T$. As a result, *conditioned on* $\{A_t\}_{t=1}^T$ and $Y_0$, $\{Y_t\}_{t=1}^T$ is a Markov chain with Gaussian transition densities:

$$k_{1:T|0,a}^{(\alpha)}(y_{1:T}|y_0, a_{1:T}) = \prod_{t=1}^T \phi_d(y_t|\gamma_t y_{t-1}, \sigma_t^2 a_t) .$$

Again, we can explicitly characterize the conditional distribution of $Y_t$ given $Y_0, \{A_t\}_{t=1}^T$. Setting for any $t \in \{1, \dots, T\}, \Sigma_{1 \to t}(a_{1:t}) := \sum_{k=1}^t (\gamma_{1 \to t} a_k^{1/2} \sigma_k / \gamma_{1 \to k})^2$, we show in Proposition 5 that:

$$Y_t \stackrel{d}{=} \gamma_{1 \to t} Y_0 + \Sigma_{1 \to t}(A_{1:t}) \varepsilon_t , \tag{6}$$

where $\varepsilon_t \sim \mathcal{S}_\alpha^i(0, \mathrm{I}_d)$. We propose approximating the backward process associated to $\{Y_t\}_{t=0}^T$, *given* $\{A_t\}_{t=1}^T$, adapting the DDPM approach. This time, for the backward process, we use the conditional density of $Y_{t-1}$ given $Y_0, Y_t$ and $A_{1:T}$:

$$\overleftarrow{k}_{1:T|0,a}^{(\alpha)}(y_{1:T}|y_0, a_{1:T}) := p_T^{(\alpha)}(y_T) \prod_{t=T}^2 k_{t-1|0,t,a}^{(\alpha)}(y_{t-1}|y_t, y_0, a_{1:t}),$$

where $k_{t-1|0,t,a}^{(\alpha)}(\cdot|y_t, y_0, a_{1:t})$ is now the tractable density of a Gaussian distribution:

**Proposition 1.** *The density of the backward process associated to $\{Y_t\}_{t=0}^T$ given $Y_0, \{A_t\}_{t=1}^T$, denoted by $k_{t-1|t,0,a}^{(\alpha)}(\cdot|y_t, y_0, a_{1:t})$, is the density of a Gaussian distribution $\mathcal{N}(\tilde{\mathrm{m}}_{t-1}, \tilde{\Sigma}_{t-1})$, with mean $\tilde{\mathrm{m}}_{t-1}$ and variance $\tilde{\Sigma}_{t-1}$ equal to:*

$$\tilde{\mathrm{m}}_{t-1}(y_t, y_0, a_{1:t}) = \frac{1}{\gamma_t} (y_t - \Gamma_t(a_{1:t})\sigma_{1 \to t}\epsilon_t(y_t, y_0)) , \quad \tilde{\Sigma}_{t-1}(a_{1:t}) = \Gamma_t(a_{1:t})\Sigma_{1 \to t-1}(a_{1:t-1}) ,$$

*where $\Sigma_{1 \to t}$ is as in* (6), $\Gamma_t = 1 - \gamma_t \Sigma_{1 \to t-1} / \Sigma_{1 \to t}$, *and* $\epsilon_t(y_t, y_0) = (y_t - \gamma_{1 \to t} y_0)/\sigma_{1 \to t}$.

See Appendix C.3 for the derivations required for Proposition 1. For our generative model, we reconsider the family of Gaussian variational approximation introduced in (4), modified to account for an iid. sequence $\{A_t\}_{t=1}^T$: $\overleftarrow{q}_{0:T}^\theta(y_{0:T}) := \int \overleftarrow{q}_T^\theta(y_T) \prod_{t=T}^1 \overleftarrow{q}_{t-1|t,a}^\theta(x_{t-1}|x_t, a_{1:t})\psi_{1:T}^{(\alpha)}(a_{1:T})\mathrm{d}a_{1:T}$, where $\psi_{1:T}^{(\alpha)}(a_{1:T}) = \prod_{t=1}^T \psi^{(\alpha)}(a_t)$ and $\psi^{(\alpha)}$ denotes the density of $\mathcal{S}_{\alpha/2,1}(0, c_A)$, and

$$\overleftarrow{q}_{t-1|t,a}^\theta(y_{t-1}|y_t, a_{1:t}) = \phi_d(y_{t-1}|\hat{\mathrm{m}}_{t-1}^\theta(y_t, a_{1:t}), \hat{\Sigma}_{t-1}^\theta(y_t, a_{1:t})) .$$

### 3.3 VARIATIONAL INFERENCE OBJECTIVE

We consider the following loss function:

$$\mathscr{L}^{\mathrm{L}}(\theta) := \mathbb{E}\left[\sum_{t=2}^T \left(\mathscr{L}_{t-1}^{\mathrm{L}}(\theta, A_{1:T})\right)^r\right], \qquad \text{where} \tag{7}$$

$$\mathscr{L}_{t-1}^{\mathrm{L}}(\theta, A_{1:t}) := \mathbb{E}\left[\mathrm{KL}\left(k_{t-1|t,0,a}^{(\alpha)}(\cdot|Y_t, Y_0, A_{1:t}) \,\|\, \overleftarrow{q}_{t-1|t,a}^\theta(\cdot|Y_t, A_{1:t})\right)\Big|A_{1:t}\right] ,$$

and $r > 0$, $k_{t-1|0,t,A}^{(\alpha)}$ denotes the conditional density of $Y_{t-1}$ given $Y_0, Y_t$ and $A_{1:T}$. In order to ensure that the expectations with respect to $A_{1:T}$ are finite, we need to choose $r < \frac{\alpha}{2}$ when $\alpha \in (1, 2)$. For simplicity, in the rest of the paper, we will use $r = \frac{1}{2}$.

A comparison with DDPM's loss function $\mathscr{L}^{\mathrm{D}}$ (cf. (3)) immediately illustrates that, thanks to Theorem 1, $\mathscr{L}^{\mathrm{L}}$ is almost identical to $\mathscr{L}^{\mathrm{D}}$ up to taking expectations with respect to one-dimensional random variables and the taking square-root of the summands[4]. We show in Appendix C.4.3 that, alike DDPM, our loss is obtained from a KL minimization principle, serving as an upper bound to:

$$\mathbb{E}\left[\left(\mathrm{KL}(k_{0|a}^{(\alpha)}(\cdot|A_{1:T})|\overleftarrow{q}_{0|a}^\theta(\cdot|A_{1:T}))\right)^r\right] ,$$

---

[4]We note that, with the choices of $\alpha = 2$ and $r = 1$, we exactly recover DDPM.

when $r \in (0, 1]$. This additionally shows how any zero of the loss corresponds to a perfect generative model, while maintaining a similar objective function. The crucial property of (7) is that, since both $k_{t-1|t,0,a}^{(\alpha)}$ and $\overleftarrow{q}_{t-1|t,a}^{\theta}$ are Gaussian (thanks to the conditioning), the KL term admits a closed-form analytical formula, as in the case of DDPM.

From a practical perspective, (7) suggests that we can use the same software architecture as for DDPM, with a slight modification to compute the outer expectation, which can be simply estimated by a Monte Carlo, or median-of-means procedure (Lugosi & Mendelson (2019)). In order to obtain a final denoising training loss, we provide three design choices, for which the full details are given in Appendix C.4. They are similar to what is classically done in diffusion models (Ho et al. (2020); Nichol & Dhariwal (2021); Karras et al. (2022)):

**D1.** We set a fixed variance $\hat{\Sigma}_t^{\theta} = \tilde{\Sigma}_t$ for the reverse process.

**D2.** We reparameterize the model to predict the value of $\epsilon_t(y_t, y_0)$ with a network $\hat{\epsilon}_t^{\theta}$, setting

$$\hat{\mathfrak{m}}_{t-1}^{\theta}(Y_t, A_{1:t}) = \frac{1}{\gamma_t}\left(Y_t - \sigma_{1 \to t}\Gamma_t(A_{1:t})\hat{\epsilon}_t^{\theta}(Y_t, A_{1:t})\right) .$$

Moreover, we drop the dependency of $\hat{\epsilon}_t^{\theta}$ on $\{A_t\}_{t=1}^{T}$, making $\hat{\epsilon}_t^{\theta}$ only a function of $Y_t$. This enables re-using classical diffusion models network architectures.

**D3.** Assuming D1, D2, we obtain $\mathscr{L}_{t-1}^{\mathrm{L}}(\theta) = \mathbb{E}\left[\lambda_{t,A_{1:t}}^2 \|\hat{\epsilon}_t^{\theta}(Y_t, A_{1:t}) - \epsilon_t(Y_t, Y_0)\|^2\right]$ where $\lambda_{t,a_{1:t}} = \Gamma_t(a_{1:t})\sigma_{1 \to t}/2\gamma_t\tilde{\Sigma}_t$, and $\epsilon_t(y_t, y_0) = (y_t - \gamma_{1 \to t}y_0)/\sigma_{1 \to t}$. We then fix $\lambda_{t,a_{1:t}} = 1$.

**Proposition 2** (Simplified denoising loss)**.** *With the design choices D1, D2, D3, we obtain the simplified denoising objective function:*

$$\mathscr{L}^{\mathrm{Simple}}(\theta) = \sum_{t=1}^{T} \mathbb{E}\left[\mathbb{E}\left(\|\hat{\epsilon}_t^{\theta}(Y_t) - \epsilon_t(Y_t, Y_0)\|^2 \mid A_{1:t}\right)^{1/2}\right] , \tag{8}$$

*where the model $\hat{\epsilon}_t^{\theta}$ is designed to fit the noise $\epsilon_t(Y_t, Y_0) = (Y_t - \gamma_{1 \to t}Y_0)/\sigma_{1 \to t}$ added at time-step $t$, and $\gamma_{1 \to t}, \sigma_{1 \to t}$ are as given in Proposition 4. Thus the model is not learning a heavy-tailed distribution directly, but the light-tailed conditional distribution.*

See Appendix C.4.2 for the derivations required for Proposition 2. Finally, in Appendix C.5, we show that on some conditions, satisfied under design choices D1, D2, D3, the expectation of each term in $\mathscr{L}^{\mathrm{L}}$ can be rewritten as an expectation with respect to only one univariate random variable (as opposed to $t$ variables, i.e., $A_{1:t}$), reducing the additional computational burden of accommodating heavy tails. As we will estimate the expectations by Monte Carlo averaging, reducing the number of random variables in the expectation is equally important to reduce the error in the estimation. The resulting loss is given in Proposition 9.

### 3.4 DETERMINISTIC SAMPLING WITH DLIM

Using the same techniques as in denoising diffusion implicit models (DDIM, Song et al. (2020)), we can recover a deterministic sampling scheme. We call this algorithmic extension Denoising Lévy Implicit Models (DLIM), which details are given in Appendix D. We recapitulate the resulting sampling and training algorithms in Appendix A. In this context and in the case of the non-isotropic Cauchy distribution ($\alpha = 1$), it is possible to bypass the data augmentation technique, since a closed-form KL divergence between two such distributions exists. Full details about these Cauchy DLIM are given in Appendix D.4, but we leave their experimental exploration to future work.

### 3.5 COMPARING DLPM TO LIM

The objective function (7) is slightly different from the one obtained in the continuous setting of LIM by Yoon et al. (2023). The training equations are very similar, and can be reformulated to involve a denoising loss (see Appendix E.1, and (29)):

$$\mathcal{L}_{t-1} : \theta \mapsto \mathbb{E}\left(\|\hat{\epsilon}_t^{\theta}(X_t) - \epsilon_t(X_t, X_0)\|_p^{\eta}\right) .$$

As a refresher, in the case of DDPM, one sets $p = 2, \eta = 2$. In the case of DLPM, our discussion leads us to the choice $p = 2, \eta = 1$ (see (8)). In the case of LIM, the theory relies on a squared

$\ell_2$ loss, setting $p = 2, \eta = 2$ in order to properly derive the loss and effectively approximate the true score of the data, at various noisescales. One must therefore make the assumption that $\mathcal{L}_{t-1}$ is not infinite for each parameter $\theta$ considered, which may not hold since $\epsilon_t(X_t, X_0)$ is $\alpha$-stable distributed and admits no variance. In the experiments for LIM, one is forced to revert to an $\ell_1$ loss, by setting $\eta = 1, p = 1$, to obtain a stable training, potentially indicating that the loss is infinite.

Additionally, DLPM and LIM yield different backward processes, which in turn lead to distinct sampling algorithms – cf. Table 4 in the Appendix. Finally, the LIM framework can only accommodate isotropic noise. We refer to Appendix E.1 for a detailed comparison between DLPM and LIM.

## 4 EXPERIMENTS

After the groundwork of the previous sections, we design experiments to demonstrate the practical strengths of our DLPM approach as compared to LIM, apart from its technical simplicity. We recall that setting $\alpha = 2$ simply reverts LIM to classical diffusion, and DLPM reverts to DDPM, apart from the square-root in the loss function. As specified in (Yoon et al. (2023), Appendix G), LIM relies on gradient and noise clipping, which introduces extra hyperparameters that must be fine-tuned for each dataset. In the experiments, we use these clipping parameters only when specified. The experimental details relative to this section are available in Appendix G.

As our loss function (7) involves an expectation with respect to $A_{1:T}$, we propose estimating it by using the *median-of-means* estimator, which is known to have better performance for heavy-tailed distributions (Lugosi & Mendelson (2019)). For an integer $M$, this approach requires sampling $M^2$ many $A_{1:T}$ terms, then split them into $M$ groups of size $M$. To approximate the expectation, we take the sample mean of each group, and finally take the median of the computed $M$ sample means. In our experiments, we explore $M = 1$ (approximating the expectation with only one sample), denoted simply by DLPM, and $M = 5$, denoted by DLPM$_5$. Similarly, we denote DLIM and DLIM$_5$ for the corresponding deterministic sampling schemes.

Finally, we consider the range $1.5 \leqslant \alpha \leqslant 2.0$. In our experiments on images, we make use of the dataset CIFAR10_LT (long tail), that has been introduced in Yoon et al. (2023) as an unbalanced modification of the CIFAR10 dataset.

### 4.1 DATA COVERAGE AND MODE COLLAPSE IN TWO-DIMENSIONAL DATA

Before progressing to higher dimensional problems, we start with easily controlled and visualized two-dimensional datasets, in order to validate the competitiveness of our method in the contexts where heavy-tailed diffusions are of interest. In particular, we consider heavy-tailed and unbalanced multi-modal datasets. See Appendix G.1 for details about the experimental setup in these settings.

**Enhancing data coverage: capturing the tail of the distribution.** We set $d = 2$ and generate data points distributed as $\mathcal{S}_\alpha^i(0, 0.05 \cdot I_d)$, with $\alpha = 1.7$. Our aim is to test the ability of each method to cover the dataset correctly; the main challenge is to correctly capture the tails. As we can visually observe in Figure 2, the backward diffusion process in the Gaussian case cannot produce truly heavy-tailed data, mainly stemming from the fact that its variance is always finite. As expected, we see improvements when using noise with $\alpha < 2$.

Figure 2: DLPM with $\alpha = 1.7$ and $\alpha = 2$.. The lighter-tailed process fails to capture the distribution's tail.

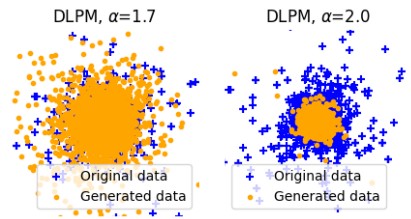

To quantify this behaviour, we utilize the mean square logarithmic error (MSLE) metric, which compares the relative error between the higher quantiles (*i.e.*, the quantiles corresponding to the tail region that has probability $1 - \xi$) of the true and the generated data distribution (see Appendix G.3 for detailed definitions). We observe in Table 1 how, as $\alpha$ gets smaller, one gets better tail approximation. Furthermore, DLPM consistently outperforms LIM, indicating that the generation process benefits from the heavy-tailed denoising formulation, rather than the continuous-time one, in this setting.

| Method | $\alpha = 1.5$ | $\alpha = 1.6$ | $\alpha = 1.7$ | $\alpha = 1.8$ | $\alpha = 1.9$ | $\alpha = 2.0$ |
|---|---|---|---|---|---|---|
| DLPM | **0.160** ± 0.128 | **0.081** ± 0.078 | **0.071** ± 0.028 | **0.099** ± 0.044 | **0.132** ± 0.101 | 0.798 ± 0.601 |
| DDPM | - | - | - | - | - | 0.528 ± 0.400 |
| | | | | | | *1.0e-1* |
| LIM | 0.743 ± 0.290 | 0.497 ± 0.311 | 0.267 ± 0.077 | 0.653 ± 0.413 | 2.444 ± 1.067 | 1.239 ± 0.240 |
| | *1.0e-08* | *8.6e-06* | *1.3e-10* | *8.8e-06* | *7.9e-09* | *5.0e-3* |

Table 1: $\text{MSLE}_{\xi=0.95}$ ↓ averaged over 20 runs. Figures below scores corresponds to $p$-values from Welch's $t$-test (assuming unequal variances), comparing the mean of DLPM with the given method.

**Enhancing data coverage: addressing mode collapse.** To assess the robustness of DLPM to mode collapse, we consider an unbalanced mixture of nine Gaussian distributions. We set their standard deviation to 0.05 and arrange them in a grid-like pattern with equal spacing, in the square $[-1, 1]^2$. Their mixture weights range from .01 to .3 [5]. We use the $F_1^{\text{pr}}$ score (*i.e.*, the harmonic mean of precision and recall, see Appendix G.3) to assess, in a single summary statistic, the quality and diversity in the generated data. As shown in Table 2, we are able to achieve improved scores by choosing a tail index $\alpha < 2$ with DLPM. This is not necessarily the case for LIM, which is consistently outperformed. Finally, DLPM$_5$ shows its strengths with better performance over all the range of $\alpha$, though at the cost of 5 times the run time.

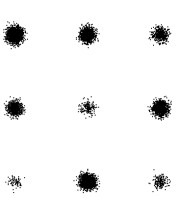

Figure 3: Gaussian grid

| Method | $\alpha = 1.5$ | $\alpha = 1.6$ | $\alpha = 1.7$ | $\alpha = 1.8$ | $\alpha = 1.9$ | $\alpha = 2.0$ |
|---|---|---|---|---|---|---|
| DLPM | 0.933 ± 0.018 | 0.923 ± 0.005 | 0.933 ± 0.028 | 0.923 ± 0.024 | 0.907 ± 0.034 | 0.862 ± 0.028 |
| DLPM$_5$ | **0.944 ± 0.013** | **0.943 ± 0.021** | **0.943 ± 0.010** | **0.941 ± 0.014** | **0.928 ± 0.016** | - |
| | *9.0e-3* | *1.6e-05* | *7.4e-2* | *9.0e-4* | *3.9e-3* | |
| LIM | 0.842 ± 0.039 | 0.850 ± 0.046 | 0.868 ± 0.034 | 0.874 ± 0.030 | 0.884 ± 0.017 | 0.874 ± 0.027 |
| | *1.7e-14* | *1.3e-09* | *5.7e-11* | *3.9e-09* | *1.9e-3* | *9.6e-2* |
| DDPM | - | - | - | - | - | 0.867 ± 0.029 |
| | | | | | | *5.0e-1* |

Table 2: $F_1^{\text{pr}}$ ↑ score, averaged over 30 runs. Figures below scores corresponds to $p$-values from Welch's $t$-test (assuming unequal variances), comparing the mean of DLPM with the given method.

## 4.2 EXPERIMENTS ON IMAGE DATA

To fairly illustrate the differences between LIM and DLPM, we use the same improved DDPM neural network architecture, as designed in Nichol & Dhariwal (2021). The specific configuration for each dataset is carefully described in Appendix G.2. Our experiments are designed to compare deterministic and stochastic generation methods under varying conditions. As a visual check, examples of generated images are listed in Appendix G.4.

**Convergence speed.** Consistent with existing literature Song et al. (2020), our findings as shown in Figure 4 confirm that deterministic generation outperforms its stochastic counterpart significantly, especially when fewer than 100 diffusion steps are used, on both MNIST and CIFAR10_LT. As the number of diffusion steps increases, both of these sampling methods produce similar results. This observation highlights that the advantages of the diffusion process do not only stem from increased randomness at sampling time. These heavy-tailed processes may define more appropriate vector field on which the noise is transported back to the original data distribution, which would lead to improved model performance (see Karras et al. (2022) for similar discussions on DDPM vs DDIM).

The previous observations are quantitatively supported in Table 3, where we present results for both deterministic and stochastic sampling strategies. We compare both methods on stochastic generation at a high step count, to compare their performance at their best regime, and on deterministic generation at a small step count, to assess the tradeoff in computations/quality offered by both methods. As we can see, DLPM surpasses LIM on both datasets. Moreover, these results show that LIM's performance deteriorates significantly when clipping is not used, raising questions about whether

---

[5]The exact mixture weights are $\{.01, .02, .02, .05, .1, .15, .15, .2, .3\}$.

Table 3: FID↓, 1000 sampling steps for LIM and DLPM, 25 sampling steps for LIM-ODE and DLIM.

Figure 4: FID↓, $\alpha = 1.7$

| MNIST | $\alpha = 1.5$ | $\alpha = 1.7$ | $\alpha = 1.8$ | $\alpha = 1.9$ | $\alpha = 2.0$ |
|---|---|---|---|---|---|
| DDPM | - | - | - | - | **3.43** |
| LIM | 14.37 | 11.54 | 11.18 | 13.75 | 11.69 |
| *w/ clipping* | *4.08* | *5.17* | *6.81* | *11.20* | |
| DLPM$_5$ | **3.80** | 3.03 | **2.51** | **2.71** | - |
| DLPM | 5.39 | **2.94** | 2.93 | 3.24 | 3.63 |
| DDIM | - | - | - | - | **5.16** |
| LIM-ODE | 49.63 | 78.59 | 92.93 | 109.48 | 29.04 |
| *w/ clipping* | *45.72* | *68.15* | *85.09* | *113.20* | |
| DLIM$_5$ | **3.37** | 2.93 | 3.44 | 4.31 | - |
| DLIM | 3.38 | **2.81** | **3.18** | **3.27** | 5.18 |
| CIFAR10_LT | | | | | |
| DDPM | - | - | - | - | **19.05** |
| LIM | 75.38 | 35.15 | 31.14 | 21.68 | 21.56 |
| *w/ clipping* | *16.13* | *16.21* | *17.67* | *19.24* | |
| DLPM | **16.10** | **18.00** | **19.94** | **20.21** | 21.07 |
| DDIM | - | - | - | - | **23.44** |
| LIM-ODE | 42.07 | 91.64 | 105.95 | 407.79 | 32.00 |
| *w/ clipping* | *30.17* | *65.78* | *84.55* | *101.70* | |
| DLIM | **20.69** | **20.77** | **21.96** | **22.79** | 23.99 |

**DLIM**, 5 steps  10 steps  25 steps  **LIM**, 5 steps  10 steps  25 steps

Figure 5: DLIM and LIM-ODE with small number of steps, on the Gaussian grid of Figure 3.

the framework of LIM is inherently well-suited for heavy-tailed distributions. More interestingly, we observe in Table 3 that DLPM consistently outperforms LIM and offers satisfying image quality at low number of steps, both for stochastic and deterministic sampling.

Generated images after 25 steps achieve a FID score of $2.81$ on MNIST and of $20.69$ on CIFAR10_LT, as compared to respectively $45.72$ and $30.17$ for LIM-ODE with clipping. On MNIST, with $\alpha = 1.7$, DLIM is able to match the sample quality of DLPM with 40 times less diffusion steps, further proving its efficacy. To visualize these behaviours, we display on Figure 5 different generation with varying time horizon $T$. We can see how the backward process defined by DLIM is able to approach the true data distribution more accurately.

Eventhough lower $\alpha$ usually entails lower FID in this table, LIM-ODE shows worse performance than Gaussian diffusion at 25 reverse steps; since the image quality is still monotonically decreasing with $\alpha$, except for $\alpha = 2$, we can conjecture that the initial instability introduced by heavy-tails are quickly counterbalanced by their benefits. DLPM$_5$ shows consistent improvement over baseline, more particularly for stochastic sampling. We provide results for non-isotropic generation, and additional metrics on image data in Appendix G.4, further supporting our claims.

## 5 CONCLUSION

In this study, we proposed DLPM and DLIM, as heavy-tailed generalizations of DDPM and DDIM. Contrary to similar methods, we believe our approach will be more accessible to the community, thanks to its elementary tools. The various experiments conducted suggest DLPM is more effective in leveraging the characteristics of heavy-tailed distributions, providing robust performance across heavy-tailed data, unbalanced datasets and requiring a lower number of diffusion steps.

ACKNOWLEDGMENT

A.D is funded by the European Union (ERC, Ocean, 101071601). D.S and U.S are partially funded by the European Union (ERC, Dynasty, 101039676). Views and opinions expressed are however those of the author(s) only and do not necessarily reflect those of the European Union or the European Research Council Executive Agency. Neither the European Union nor the granting authority can be held responsible for them.

U.S is additionally funded by the French government under management of Agence Nationale de la Recherche as part of the "France 2030" program, reference ANR-23-IACL-0008 (PR[AI]RIE-PSAI).

The authors are grateful to the CLEPS infrastructure from the Inria of Paris for providing resources and support.

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

APPENDIX

The Appendix is organized as follows:

- In Appendix A, we provide the pseudo-code for the training and sampling algorithms of DLPM and DLIM.

- In Appendix B, we characterize the stability property of the $\alpha$-stable distribution, and we give explicit formulas for the distribution of the sum of two $\alpha$-stable random variables.

- In Appendix C, we provide the detailed theory and derivations of the DLPM framework. Working on the process $\{X_t\}_{t=0}^T$ defined in (2), and its data augmentation counterpart $\{Y_t\}_{t=0}^T$ defined in (5), we start in Appendix C.2 by characterizing the distribution of $X_t$ given $X_0$ from a given noise schedule $\{(\gamma_t, \sigma_t)\}_{t=1}^T$, as this defines the characteristic location $\gamma_{1 \to t}$ and scale $\sigma_{1 \to t}$ of the process at time $t$. Likewise we characterize the distribution of $Y_t$ given $Y_0, \{A_t\}_{t=1}^T$, as this defines the characteristic location $\gamma_{1 \to t}$ and variance $\Sigma_{1 \to t}$ of the augmented process at time $t$.

  In Appendix C.3, we focus on the Gaussian trick exploited by the process $\{Y_t\}_{t=0}^T$ defined in (5). This leads us to the an explicit formula for the Gaussian density of the backward process conditioned on a sequence $\{A_t\}_{t=1}^T$ of $\alpha$-stable random variables.

  In Appendix C.4, we further put this characterization into good use by obtaining a closed-form formula for the loss function . Follows a discussion on design choices for the model and the loss function at hand, which leads us to the simplified training loss (8) corresponding to the denoising loss for $\alpha$-stable diffusion.

  In Appendix C.4.3 we provide a more principled approach to derive the loss function at hand.

  Finally, in Appendix C.5, we provide a faster sampling strategy for training DLPM, computing each loss term $\mathscr{L}_{t-1}^{\mathrm{L}}$ using only two heavy-tailed random variables per datapoint, instead of $t$ random variables.

- In Appendix D, we adapt the setting for deterministic sampling in classical denoising diffusion DDIM (Song et al. (2020)) to our $\alpha$-stable framework. We naturally call this extension DLIM. Alike DDIM, it is true that the same neural networks can be used for both the DLIM and DLPM generation procedures.

- In Appendix E, we compare in more details the two discrete and continuous frameworks DLPM and LIM, underlining how they offer two distinct loss functions, training and sampling algorithms.

- In Appendix F, we give proofs relative to our technique for faster training, as introduced in Appendix C.5.

- In Appendix G, we provide additional experimental details.

## A   ALGORITHMS FOR DLPM AND DLIM

In this section, we explicitly provide the algorithms needed to train and sample from the DLPM and DLIM generative methods.

### A.1   TRAINING

The same models can be shared between DLPM and DLIM, as underlined in Appendix D.3. We introduce the values $\sigma_{1 \to t}, \gamma_{1 \to t}$ determined by the noise schedule, as presented in Proposition 4:

$$\gamma_{1 \to t} = \prod_{i=1}^t \gamma_t \,, \qquad \sigma_{1 \to t} = \left[ \sum_{i=1}^t \left( \frac{\gamma_{1 \to t}}{\gamma_{1 \to i}} \sigma_i \right)^\alpha \right]^{1/\alpha} \,.$$

We define $c_A = \cos^{2/\alpha}(\pi\alpha/4)$ as in Theorem 1. We make the design choices D1, D2, D3 for our model, as described in Appendix C.4.2. Finally, we use the method for faster sampling as described in Appendix C.5, Proposition 9. The resulting training algorithm is given in Algorithm 2.

---

**Algorithm 2** DLPM training - simplified loss

---

**Require:** model $\{\hat{\epsilon}_t^\theta\}_{t=1}^T$, noise schedule $\{(\gamma_t, \sigma_t)\}_{t=1}^T$, data $Y_0$
1: Sample $t \sim \mathcal{U}[1, T]$
2: Sample $\bar{A}_t \sim \mathcal{S}_{\alpha/2,1}(0, c_A)$
3: Sample $G_t \sim \mathcal{N}(0, \mathrm{I}_d)$
4: $Y_t \leftarrow \gamma_{1 \to t} Y_0 + \sigma_{1 \to t} \bar{A}_t^{1/2} G_t$
5: $L_{t-1} \leftarrow \|\hat{\epsilon}_t^\theta(Y_t) - \bar{A}_t^{1/2} G_t\|_2$
     **return** $L_{t-1}$

---

## A.2 SAMPLING

Given the noise schedule and a sequence $\{A_t\}_{t=1}^T$ of $\alpha$-stable random variables we define:

$$\Sigma_{1 \to t}(A_{1:t}) = \sum_{k=1}^t \left( \frac{\gamma_{1 \to t}}{\gamma_{1 \to k}} \sqrt{A_k} \sigma_k \right)^2 , \quad \Gamma_t(A_{1:t}) = 1 - \frac{\gamma_t^2 \Sigma_{1 \to t-1}(A_{1:t-1})}{\Sigma_{1 \to t}(A_{1:t})} , \qquad (9)$$

see Proposition 6 and Equation (12) from Proposition 5 for precise statements. We give our sampling algorithm for DLPM in Algorithm 3.

---

**Algorithm 3** Stochastic sampling (DLPM)

---

**Require:** model $\{\hat{\epsilon}_t^\theta\}_{t=1}^T$, noise schedule $\{(\gamma_t, \sigma_t)\}_{t=1}^T$
1: Sample $Y_T \sim \mathcal{S}(0, \sigma_{1 \to T} \mathrm{I}_d)$
2: Sample $\{A_t\}_{t=1}^T$ i.i.d., $A_t \sim \mathcal{S}_{\alpha/2,1}(0, c_A)$
3: **for** $t \leftarrow T$ to 1 **do**
4:     Compute $\Sigma_{1 \to t}, \Sigma_{1 \to t-1}, \Gamma_t$ as in (9)
5:     Sample $G_t \sim \mathcal{N}(0, \mathrm{I}_d)$
6:     $\hat{\Sigma}_{t-1}^\theta \leftarrow \Gamma_t \Sigma_{1 \to t-1}$
7:     $Y_{t-1} \leftarrow \dfrac{Y_t}{\gamma_t} - \Gamma_t \sigma_{1 \to t} \hat{\epsilon}_t^\theta(Y_t) + \sqrt{\hat{\Sigma}_{t-1}^\theta} G_t$
8: **end for**
     **return** $Y_0$

---

For DLIM, one can potentially provide $Y_T$ as input, in order to use the support of the noise distribution as a latent space, alike what is done by Song et al. (2020). We give our DLIM sampling algorithm in Algorithm 4.

---

**Algorithm 4** Deterministic sampling (DLIM)

---

**Require:** model $\{\hat{\epsilon}_t^\theta\}_{t=1}^T$, noise schedule $\{(\gamma_t, \sigma_t)\}_{t=1}^T$
1: Sample $Y_T \sim \mathcal{S}_\alpha^{\mathrm{i}}(0, \mathrm{I}_d)$
2: **for** $t \leftarrow T$ to 1 **do**
3:     $Y_{t-1} \leftarrow \dfrac{Y_t}{\gamma_t} - \left( \dfrac{\sigma_{1 \to t}}{\gamma_t} - \sigma_{1 \to t-1} \right) \hat{\epsilon}_t^\theta(Y_t)$
4: **end for**
     **return** $Y_0$

---

## B ADDITIONAL REMARK ON $\alpha$-STABLE DISTRIBUTIONS

Stable distributions are closed under convolution for a fixed value of $\alpha$. Since convolution is equivalent to multiplication of the Fourier-transformed function, it follows that the product of two stable characteristic functions with the same $\alpha$ will yield another such characteristic function. We precisely characterize this stability property in the following proposition:

**Proposition 3** (See (Nolan, 2020, Proposition 1.3)). *Let $X_1, X_2$ be two random variables respectively distributed as $X_1 \sim \mathcal{S}_{\alpha,\beta_1}(\mu_1, \sigma_1)$ and $X_2 \sim \mathcal{S}_{\alpha,\beta_2}(\mu_2, \sigma_2)$, with $\mu_1, \mu_2, \beta_1, \beta_2 \in \mathbb{R}$ and $\sigma_1, \sigma_2 > 0$. Then, $X = X_1 + X_2$ is distributed as $\mathcal{S}_{\alpha,\beta}(\mu, \sigma)$ where:*

$$\mu = \mu_1 + \mu_2 \,, \qquad \sigma = (\sigma_1^\alpha + \sigma_2^\alpha)^{1/\alpha} \,, \qquad \beta = \frac{\beta_1 \sigma_1^\alpha + \beta_2 \sigma_2^\alpha}{\sigma_1^\alpha + \sigma_2^\alpha} \,.$$

In particular, when $X_1, X_2$ are such that $X_1 \sim \mathcal{S}_\alpha(0, \sigma_1), X_2 \sim \mathcal{S}_\alpha(0, \sigma_2)$, then $X = X_1 + X_2 \sim \mathcal{S}_\alpha(0, (\sigma_1^\alpha + \sigma_2^\alpha)^{1/\alpha})$, which is the key relation used in the later characterizations of the distribution of our forward process.

## C   THEORETICAL DERIVATIONS FOR DLPM

In this section, we provide the detailed theory and associated derivations of the DLPM framework.

### C.1   SETTING AND NOTATIONS

We will denote by $\phi_d(\cdot|\mu, \Sigma)$ the density of $\mathcal{N}(\mu, \Sigma)$, where $\mu \in \mathbb{R}^d$ and $\Sigma \in \mathbb{R}^{d \times d}$. We will denote by $\psi^{(\alpha)}$ the density of $\mathcal{S}_{\alpha/2,1}(0, c_A)$, where $c_A = \cos^{2/\alpha}(\pi\alpha/4)$.

**Forward process.**   We reintroduce the setting presented in Section 3, with the noising schedule being denoted by $\{(\gamma_t, \sigma_t)\}_{t=1}^T$, and the following **forward process** on which DLPM is based:

$$X_0 \sim p_\star \,, \quad \text{and} \quad X_t = \gamma_t X_{t-1} + \sigma_t \epsilon_t^{(\alpha)}, \tag{10}$$

where $p_\star$ is the data distribution and $\{\epsilon_t^{(\alpha)}\}_{t=1}^T$ are independent random variables distributed as $\mathcal{S}_\alpha^{\mathrm{i}}(0, \mathrm{I}_d)$.

**Data augmentation process.**   We also introduce the associated **data augmentation process**:

$$Y_0 \sim p_\star \,, \quad \text{and} \quad Y_t = \gamma_t Y_{t-1} + \sigma_t A_t^{1/2} G_t \,, \tag{11}$$

where $\{G_t\}_{t=1}^T$ and $\{A_t\}_{t=1}^T$ are independent random variables distributed according to $G_t \sim \mathcal{N}(0, \mathrm{I}_d)$ and $A_t \sim \mathcal{S}_{\alpha/2,1}(0, c_A)$, with $c_A = \cos^{2/\alpha}(\pi\alpha/4)$. From Theorem 1, $\{Y_t\}_{t=0}^T$ is a Markov chain that admits the same distribution as $\{X_t\}_{t=0}^T$. We will denote by $p_t^{(\alpha)}$ the distribution of $Y_t$, and by $k_{t|t-1}^{(\alpha)}(\cdot|\cdot)$ the transition density associated to the Markov chain (10).

**Backward process.**   A **backward process** associated to the Markov chain $\{Y_t\}_{t=0}^T$ is a Markov chain $\{\overleftarrow{Y}'_t\}_{t=0}^T$ such that the two processes $\{\overleftarrow{Y}'_{T-t}\}_{t=0}^T$ and $\{Y_t\}_{t=0}^T$ have the same distribution. For ease of presentation and following classical notations, we will rather consider $\{\overleftarrow{Y}_t\}_{t=0}^T$ where $\overleftarrow{Y}_t = \overleftarrow{Y}'_{T-t}$. We will denote by $\overleftarrow{k}_{t-1|t}^{(\alpha)}(\cdot|\cdot)$ the transition densities associated to the process $\{\overleftarrow{Y}_t\}_{t=0}^T$. Since the true backward process is never available to us, we will focus on an approximation induced by a variational family. We consider the process $\{\overleftarrow{Y}_t^\theta\}_{t=0}^T$ where $\overleftarrow{Y}_T^\theta$ is distributed as $\mathcal{S}(0, \sigma_{1 \to t} \mathrm{I}_d)$, and the density of the distribution of $\overleftarrow{Y}_{t-1}^\theta$ conditioned on $\overleftarrow{Y}_t^\theta$ are given by $\overleftarrow{q}_{t-1|t}^\theta(\cdot|\cdot)$, where $\theta \in \mathbb{R}^D$ parameterizes a neural network. We also denote by $\overleftarrow{q}_{0:T}^\theta$ the joint distribution of $\{\overleftarrow{Y}_t^\theta\}_{t=0}^T$ and by $\overleftarrow{q}_t^\theta$ the marginal distribution of $\overleftarrow{Y}_t^\theta$.

**Further notations.**   Finally, we denote by $p_t^\alpha(\cdot|a_{1:t})$, $k_{t|t-1,a}^{(\alpha)}(\cdot|\cdot, a_{1:t})$, and $\overleftarrow{q}_{t-1|t,a}^\theta(\cdot|\cdot, a_{1:t})$ the densities/transition densities associated to the processes $\{Y_t\}_{t=0}^T, \{\overleftarrow{Y}_t^\theta\}_{t=0}^T$ given $A_t = a_t$ for $1 \leqslant t \leqslant T$. We we will also write $p_t^\alpha(\cdot|y_0, a_{1:t})$, $k_{t|t-1,0,a}^{(\alpha)}(\cdot|\cdot, y_0, a_{1:t})$, and $\overleftarrow{q}_{t-1|t,0,a}^\theta(\cdot|\cdot, y_0, a_{1:t})$ when further conditioning on $Y_0$.

## C.2 FORWARD PROCESS

Let us now characterize the distribution of $X_t$ given $X_0$, and $Y_t$ given $Y_0, \{A_t\}_{t=1}^T$, which are tractable thanks to Proposition 3. These will come in handy, for instance when working on the backward process in Appendix C.3.

**Proposition 4** (Distribution of $X_t$ given $X_0$)**.** *Let $\{X_t\}_{t=0}^T$ be the forward process as given in* (10)*, and $\{(\gamma_t, \sigma_t)\}_{t=1}^T$ the noise schedule. Then the distribution of $X_t$ given $X_0$ is given for any $t$ by*

$$X_t \overset{d}{=} \gamma_{1 \to t} X_0 + \sigma_{1 \to t} \bar{\epsilon}_t$$

*where $\bar{\epsilon}_t \sim \mathcal{S}_\alpha^i(0, \mathrm{I}_d)$, and $\gamma_{1 \to t}, \sigma_{1 \to t}$ are given by:*

$$\gamma_{1 \to t} = \prod_{k=1}^t \gamma_k \,, \qquad \sigma_{1 \to t} = \left( \sum_{k=1}^t \left( \frac{\gamma_{1 \to t}}{\gamma_{1 \to k}} \sigma_k \right)^\alpha \right)^{1/\alpha} \,.$$

The proof is an elementary induction based on Proposition 3.

**Proposition 5** (Distribution of $Y_t$ given $Y_0, \{A_t\}_{t=1}^T$)**.** *Let $\{Y_t\}_{t=0}^T$ be the forward process as given in* (11)*, $\{(\gamma_t, \sigma_t)\}_{t=1}^T$ the noise schedule, and $\{A_t\}_{t=1}^T$ the associated $\alpha/2$-stable random variables, parameterizing the variance of the Gaussian noise increments. Then the distribution of $Y_t$ given $Y_0, \{A_t\}_{t=1}^T$ is the Gaussian distribution with mean $\gamma_{1 \to t} Y_0$ and covariance matrix $\Sigma_{1 \to t} \mathrm{I}_d$, i.e.,*

$$Y_t \overset{d}{=} \gamma_{1 \to t} Y_0 + \Sigma_{1 \to t}(A_{1:t})^{1/2} \bar{G}_t \,,$$

*where $\bar{G}_t \sim \mathcal{N}(0, \mathrm{I}_d)$, and*

$$\gamma_{1 \to t} = \prod_{k=1}^t \gamma_k \,, \qquad \Sigma_{1 \to t}(a_{1:t}) = \sum_{k=1}^t \left( \frac{\gamma_{1 \to t}}{\gamma_{1 \to k}} \sqrt{a_k} \sigma_k \right)^2 \,. \tag{12}$$

The proof is elementary and omitted. It is worth mentioning the following recurrence, to speedup the computation of the sequence $\{\Sigma_{1 \to t}(a_{1:t})\}_{t=1}^T$:

$$\Sigma_{1 \to t}(a_{1:t}) = \sigma_t^2 a_t + \gamma_t^2 \Sigma_{1 \to t-1}(a_{1:t-1}) \,.$$

## C.3 BACKWARD PROCESS

Consider the setting of the data augmentation approach as given in (11). By the same arguments used in Section 3, it can be verified that a process starting from $p_T^{(\alpha)}$ and with transition densities $\overleftarrow{k}_{t-1|t}^{(\alpha)}(y_{t-1}|y_t) \propto p_t^{(\alpha)}(y_{t-1}) k_{t|t-1}^{(\alpha)}(y_t|y_{t-1})$ for any $y_{t-1}, y_t$ is a backward process associated with $\{Y_t\}_{t=0}^T$. However, it raises two main problems. First (i), we cannot characterize the distribution of $\overleftarrow{k}_{t-1|t}^{(\alpha)}(y_{t-1}|y_t)$, since we do not know the data distribution. Second (ii), in the case where $\alpha < 2$, we do not have access to an explicit expression for $\overleftarrow{k}_{t-1|t,0}^{(\alpha)}(y_{t-1}|y_t, y_0)$.

Regarding (i), we have access to the distribution of $Y_t$ given $Y_0$, so a valid strategy consists in devising a method relying on characterizing the backward of the process $\{Y_t\}_{t=0}^T$ given $Y_0$. This is the classical strategy used in DDPM (Ho et al. (2020)), which is possible in the case $\alpha = 2$ since $\overleftarrow{k}_{t-1|t,0}^{(\alpha)}(y_{t-1}|y_t, y_0)$ admits an analytical expression for any $y_0, y_{t-1}, y_t$, thanks to the properties of the Gaussian distribution.

Regarding (ii), in the case where $\alpha < 2$, we make use of the trick introduced in Theorem 1, justifying the data augmentation approach. We will rather characterize the density of the Markov kernels associated to the backward of the process $\{Y_t\}_{t=0}^T$ given $Y_0$ and $\{A_t\}_{t=1}^T$. This time, since we manage Gaussian noise increments, we can fall back to the classical strategy, as we further develop in the following proposition.

**Proposition 6** (Density of the backward process associated to $\{Y_t\}_{t=0}^T$ given $Y_0, \{A_t\}_{t=1}^T$)**.** *Consider the setting of the data augmentation approach as given in* (11)*. Let $\{(\gamma_t, \sigma_t)\}_{t=1}^T$ be the noise schedule at hand. Let $k_{t-1|0,t,a}^{(\alpha)}(\cdot|y_t, y_0, a_{1:t})$ be the density of the backward process associated to*

$\{Y_t\}_{t=0}^T$ *given $Y_0$ and* $\{A_t\}_{t=1}^T$*. Then* $k_{t-1|t,0,a}^{(\alpha)}(\cdot|y_t, y_0, a_{1:t})$ *is the density of a Gaussian distribution* $\mathcal{N}(\tilde{m}_{t-1}, \tilde{\Sigma}_{t-1})$ *with mean $\tilde{m}_{t-1}$ and variance $\tilde{\Sigma}_{t-1}$ such that*

$$\tilde{m}_{t-1}(y_t, y_0, a_{1:t}) = \frac{1}{\gamma_t}\left(y_t - \Gamma_t(a_{1:t})\sigma_{1\to t}\epsilon_t(y_t, y_0)\right), \quad \tilde{\Sigma}_{t-1}(a_{1:t}) = \Gamma_t(a_{1:t})\Sigma_{1\to t-1}(a_{1:t-1}),$$

(13)

*where*

$$\Sigma_{1\to t}(a_{1:t}) = \sum_{k=1}^{t}\left(\frac{\gamma_{1\to t}}{\gamma_{1\to k}}\sqrt{a_k}\sigma_k\right)^2$$

$$\epsilon_t(y_t, y_0) = \frac{y_t - \gamma_{1\to t}y_0}{\sigma_{1\to t}}$$

$$\Gamma_t(a_{1:t}) = 1 - \frac{\gamma_t^2\Sigma_{1\to t-1}(a_{1:t-1})}{\Sigma_{1\to t}(a_{1:t})}.$$

*Eventhough $\Gamma_t$ involves multiple heavy-tailed random variables, it is nonetheless bounded:* $0 \leqslant \Gamma_t \leqslant 1$.

*Proof.* To determine $k_{t-1|t,0,a}^{(\alpha)}(\cdot|y_t, y_0, a_{1:t})$, we need to work on the joint distribution of $(Y_{t-1}, Y_t)$ conditioned on $Y_0, \{A_t\}_{t=1}^T$, which is a Gaussian vector for which classical techniques will let us derive the distribution of $Y_{t-1}$ given $Y_t$. Before doing so we need to compute $\rho_t$ the covariance of $Y_{t-1}$ and $Y_t$ given $Y_0, \{A_t\}_{t=1}^T$, which we do thanks to Proposition 5:

$$\rho_t = \text{Cov}(Y_t, Y_{t-1}|Y_0, A_{1:T}) = \gamma_t\text{Cov}(Y_{t-1}, Y_{t-1}|Y_0, A_{1:T}) = \gamma_t\Sigma_{1\to t-1}I_d.$$

Denote by $k_{t-1,t|0,a}^{(\alpha)}$ the density of $(Y_{t-1}, Y_t)$ conditioned on $Y_0, A_{1:T}$. Denote by $\phi_d(\cdot|\mu, \Sigma)$ the density of a $d$-dimensional Gaussian distribution with mean $\mu$ and covariance $\Sigma$. From the results of Proposition 5, we can write

$$k_{t-1,t|0,a}^{(\alpha)}(y_{t-1}, y_t|y_0, a_{1:t}) = \phi_d\left(\begin{pmatrix} y_{t-1} \\ y_t \end{pmatrix} \middle| \begin{pmatrix} \gamma_{1\to t-1}y_0 \\ \gamma_{1\to t}y_0 \end{pmatrix}, \begin{pmatrix} \Sigma_{1\to t-1}(a_{1:t-1})I_d & \rho_t I_d \\ \rho_t I_d & \Sigma_{1\to t}(a_{1:t})I_d \end{pmatrix}\right)$$

Then the distribution of $Y_{t-1}$ given $Y_t, Y_0, A_{1:T}$ is a Gaussian distribution $\mathcal{N}(\tilde{m}_{t-1}, \tilde{\Sigma}_{t-1})$ (Holt & Nguyen, 2023, Theorem 3) with mean $\tilde{m}_{t-1}$ and variance $\tilde{\Sigma}_{t-1}$ satisfying:

$$\tilde{m}_{t-1}(y_t, y_0, a_{1:t}) = \gamma_{1\to t-1}y_0 + \frac{\rho_t}{\Sigma_{1\to t}(a_{1:t})}\left(y_t - \gamma_{1\to t}y_0\right)$$

$$\tilde{\Sigma}_{t-1}(a_{1:t}) = \Sigma_{1\to t-1}(a_{1:t-1}) - \frac{\rho_t^2}{\Sigma_{1\to t}(a_{1:t})}.$$

By defining

$$\epsilon_t(y_t, y_0) = \frac{y_t - \gamma_{1\to t}y_0}{\sigma_{1\to t}}, \quad \Gamma_t(a_{1:t}) = 1 - \frac{\gamma_t^2\Sigma_{1\to t-1}(a_{1:t-1})}{\Sigma_{1\to t}(a_{1:t})},$$

we give the final expression for the mean $\tilde{m}_{t-1}$ and variance $\tilde{\Sigma}_{t-1}$ of the distribution of $Y_{t-1}$ given $Y_t, Y_0$ and $\{A_t\}_{t=1}^T$:

$$\tilde{m}_{t-1}(y_t, y_0, a_{1:t}) = \frac{1}{\gamma_t}\left(y_t - \sigma_{1\to t}\Gamma_t(a_{1:t})\epsilon_t(y_t, y_0)\right)$$

(14)

$$\tilde{\Sigma}_{t-1}(a_{1:t}) = \Gamma_t(A_{1:t})\Sigma_{1\to t-1}(a_{1:t-1}).$$

Since $\Gamma_t(a_{1:t}) = 1 - \gamma_t^2\Sigma_{1\to t-1}(a_{1:t-1})/\Sigma_{1\to t}(a_{1:t}) = a_t\sigma_t^2/\Sigma_{1\to t}(a_{1:t})$ and $a_t, \gamma_t, \Sigma_{1\to t}, \sigma_{1\to t} > 0$, we have $0 \leqslant \Gamma_t \leqslant 1$.

$\square$

**Case** $\alpha = 2$   As we set $\alpha = 2$, the random variables $\{A_t\}_{t=1}^T$ become deterministic, equal to 2. One can check that in this case, with the variance preserving schedule

$$\gamma_t = \sqrt{1 - \beta_t}\,, \quad \gamma_{1 \to t} = \sqrt{\alpha_t}\,, \quad \sigma_t = \sqrt{\beta_t}\,, \quad \sigma_{1 \to t} = \sqrt{1 - \alpha_t}\,,$$

then:

$$\Sigma_{1 \to t} = 2\sigma_{1 \to t}^2 = 2(1 - \alpha_t)\,.$$

Further noticing that $\gamma_t = \gamma_{1 \to t}/\gamma_{1 \to t-1}$, one computes

$$\begin{aligned}
\Gamma_t &= 1 - \frac{\sigma_{1 \to t-1}^2 \alpha_t/\alpha_{t-1}}{\sigma_{1 \to t}^2} \\
&= 1 - \frac{(1 - \alpha_{t-1})\alpha_t/\alpha_{t-1}}{1 - \alpha_t} \\
&= \frac{1 - \alpha_t/\alpha_{t-1}}{1 - \alpha_t}\,,
\end{aligned}$$

so that one recovers the famous equations made popular in the seminal DDPM paper (Ho et al., 2020, Equation 7):

$$\tilde{\mathrm{m}}_{t-1} = \frac{\sqrt{\alpha_{t-1}}}{\sqrt{\alpha_t}}\left(Y_t - \frac{1 - \alpha_t/\alpha_{t-1}}{\sqrt{1 - \alpha_t}}\epsilon_t(Y_t, Y_0)\right) \tag{15}$$

$$\tilde{\Sigma}_{t-1} = (1 - \alpha_{t-1})\frac{1 - \alpha_t/\alpha_{t-1}}{1 - \alpha_t}\,,$$

with $\epsilon_t(Y_t, Y_0) = (Y_t - \sqrt{\alpha_t}Y_0)/\sqrt{1 - \alpha_t}$.

**Model for the reverse process.**   We propose approximating the backward process associated to $\{Y_t\}_{t=0}^T$, *given* $\{A_t\}_{t=1}^T$, adapting the DDPM approach. This time, for the backward process, we characterized the conditional density of $Y_{t-1}$ given $Y_0, Y_t$ and $\{A_t\}_{t=1}^T$:

$$\overleftarrow{k}_{1:T|0,a}^{(\alpha)}(y_{1:T}|y_0, a_{1:T}) := p_T^{(\alpha)}(y_T)\prod_{t=T}^1 k_{t-1|0,t,a}^{(\alpha)}(y_{t-1}|y_t, y_0, a_{1:t})\,,$$

where $k_{t-1|0,t,a}^{(\alpha)}(\cdot|y_t, y_0, a_{1:t})$ is the tractable density of a Gaussian distribution, as we have just proved in Proposition 6. We similarly reconsider the family of Gaussian variational approximation introduced in (4), modified to account for an i.i.d. sequence $\{A_t\}_{t=1}^T$:

$$\overleftarrow{q}_{t-1|t,a}^\theta(y_{t-1}|y_t, a_{1:t}) = \phi_d(y_{t-1}|\hat{\mathrm{m}}_{t-1}^\theta(y_t, a_{1:t}), \hat{\Sigma}_{t-1}^\theta(y_t, a_{1:t}))\,, \tag{16}$$

where $\phi_d$ is the density of the multivariate Gaussian distribution, so that the overall model for the backward process is the following:

$$\overleftarrow{q}_{0:T}^\theta(y_{0:T}) = \int \psi_{1:T}^{(\alpha)}(a_{1:T})p_T(y_T)\prod_{t=1}^T \overleftarrow{q}_{t-1|t,a}^\theta(y_{t-1}|y_t, a_{1:t})\mathrm{d}a_{1:T}\,,$$

where $\psi_{1:T}^{(\alpha)}(a_{1:T}) = \prod_{i=1}^T \psi^{(\alpha)}(a_i)$, and $p_T$ is the density of $\mathcal{S}_\alpha^i(0, \sigma_{1 \to T}\mathrm{I}_d)$.

## C.4   TRAINING LOSS

In this section, we draw inspiration from DDPM (Ho et al. (2020)) to obtain a loss function admitting a closed-form formula. We further provide three design choices which lead to a simplified training loss, corresponding to the denoising loss for $\alpha$-stable diffusion.

### C.4.1   CLASSICAL LOSS FOR DDPM, $\alpha = 2$

We start by reviewing what is classically done for DDPM, *i.e.*, the case $\alpha = 2$. The variational approximation $\{\overleftarrow{q}_{0:T}^\theta : \theta \in \Theta\}$, for some parameter space $\Theta$, is designed to admit the same decomposition as $\overleftarrow{k}_{0:T}^{(2)}$, *i.e.*, $\overleftarrow{q}_{0:T}^\theta(x_{0:T}) = \overleftarrow{q}_T^\theta(x_T)\prod_{t=T}^1 \overleftarrow{q}_{t-1|t}^\theta(x_{t-1}|x_t)$, where $\overleftarrow{q}_T^\theta$ is chosen to be the density of $\mathcal{S}_\alpha^i(0, \sigma_{1 \to T}\mathrm{I}_d)$ as an approximation of $p_T^{(\alpha)}$. Then, it is trained on a classical upper bound of the KL loss $\mathscr{L}^{\mathrm{D}} : \theta \mapsto \mathrm{KL}(p_\star\|\overleftarrow{q}_0^\theta)$ between the true and the generated distribution,

which is a form of evidence lower bound loss (Ho et al., 2020, Equation 5). Thus one resorts to optimize the following sum:

$$\mathscr{L}^{\mathrm{D}}(\theta) \leqslant \mathscr{L}_T^{\mathrm{D}} + \sum_{t=2}^{T} \mathscr{L}_{t-1}^{\mathrm{D}}(\theta) + \mathscr{L}_0^{\mathrm{D}}(\theta) + C \tag{17}$$

where $C$ is a constant that does not depend on $\theta$, and

$$\mathscr{L}_T^{\mathrm{D}} = \mathbb{E}\left[\mathrm{KL}\left(k_{T|0}^{(2)}(\cdot|X_0) \,\|\, \phi_d(\cdot|0, \sigma_{1\to T}\mathrm{I}_d)\right)\right]$$

$$\mathscr{L}_0^{\mathrm{D}}(\theta) = -\mathbb{E}\left[\log\left(\overleftarrow{q}_{0|1}^{\theta}(X_0|X_1)\right)\right]$$

$$\mathscr{L}_{t-1}^{\mathrm{D}}(\theta) = \mathbb{E}\left[\mathrm{KL}\left(k_{t-1|0,t}^{(2)}(\cdot|X_0, X_t) \,\|\, \overleftarrow{q}_{t-1|t}^{\theta}(\cdot|X_t)\right)\right] \;,$$

where $\{X_t\}_{t=0}^{T}$ is the process defined in (10), and $k_{t-1|0,t}^{(2)}$ is the conditional density of $X_{t-1}$ given $X_0, X_t$. We make the following classical remarks on the terms of this loss (Sohl-Dickstein et al. (2015), Yang et al. (2024)). The term $\mathscr{L}_T^{\mathrm{D}}$ does not depend on $\theta$ but only on the chosen time horizon for the forward process, that determines the final variance of the Gaussian distribution $\mathcal{N}(0, \sigma_{1\to T}\mathrm{I}_d)$. It is neglected. The effect of optimizing the first term $\mathscr{L}_0^{\mathrm{D}}(\theta)$ is negligible too.

More importantly, for the term $\mathscr{L}_{t-1}^{\mathrm{D}}(\theta)$, when using Gaussian variational approximations, *i.e.*, as

$$\overleftarrow{q}_{t-1|t}^{\theta}(x_{t-1}|x_t) = \phi_d\left(x_{t-1}|\hat{\mathtt{m}}_{t-1}^{\theta}(x_t), \hat{\Sigma}_{t-1}^{\theta}(x_t)\right) \;,$$

where $(x, \mathtt{m}, \Sigma) \mapsto \phi_d(x|\mathtt{m}, \Sigma)$ is the $d$-dimensional density of the Gaussian distribution with mean $\mathtt{m}$ and covariance matrix $\Sigma$, $\hat{\mathtt{m}}_{t-1}^{\theta}, \hat{\Sigma}_{t-1}^{\theta}$ are some functions of $x_t$ parameterized by $\theta$, it turns out that $\mathscr{L}_{t-1}^{\mathrm{D}}$ admits a closed-form expression. For a fixed variance $\hat{\Sigma}_{t-1}^{\theta} = \tilde{\Sigma}_{t-1}$, with $\tilde{\Sigma}_{t-1}$ given in (15), one resorts to optimize a convenient $\mathbb{L}_2$ loss function:

$$\mathscr{L}_{t-1}^{\mathrm{D}}(\theta) = \lambda_t \|\tilde{\mathtt{m}}_{t-1}(x_t, x_0) - \hat{\mathtt{m}}_{t-1}^{\theta}(x_t)\|^2, \tag{18}$$

where $\lambda_t, \tilde{\mathtt{m}}_t$ depend on the noise schedule $(\gamma_t, \sigma_t)$ and $x_t, x_0$.

Unfortunately, as we mentioned in Section 3.2.2, this solution cannot be used as such to learn the backward transitions associated to $\{X_t\}_{t=0}^{T}$ for $\alpha < 2$. The main issue that we face stems from the fact that the density of $\alpha$-stable distributions are in most cases unknown, in contrast to Gaussian distributions. As a result, the conditional density $x_{t-1}, x_0, x_t \mapsto k_{t-1|0,t}^{(\alpha)}(x_{t-1}|x_0, x_t)$ is unknown for $\alpha < 2$, which prevents us to have an explicit expression for $\theta \mapsto \mathscr{L}_{t-1}^{\mathrm{D}}(\theta)$.

Moreover, the absence of a second order moment for $\alpha$-stable distributions challenges the most straightforward adaptation we can make to the previous loss considering the data augmentation setting. Indeed, to fit $\theta$ to the data distribution, we aim to rely on Kullback-Leibler minimization, a.k.a. the maximum likelihood principle, and some associated upper bounds. A naive solution would consist in considering the bounds obtained applying the Jensen inequality:

$$\mathrm{KL}(p_\star \| \overleftarrow{q}_0^{\theta}) \leqslant \mathbb{E}\left[\mathrm{KL}\left[p_\star(\cdot) \| \overleftarrow{q}_{0|a}^{\theta}(\cdot|A_{1:T})\right]\right] \;,$$

and fall back to the expression obtained in (17), only with conditioning on $\{A_t\}_{t=1}^{T}$. However, as we see in (18), this expression would involve taking expectation of $A_t$, while it is distributed as $\mathcal{S}_{\alpha/2, 1}(0, c_A)$ and admits no first order moment. We are not aware of any bounds on $\mathrm{KL}(p_\star \| \overleftarrow{q}_{0|a}^{\theta}(\cdot|A_{1:T}))$ that would lead to a meaningful optimization problem due to the heavy tailed nature of the distribution of $\{A_t\}_{t=1}^{T}$.

### C.4.2 Loss for DLPM, for any $\alpha$

We now expose our methodology to address this limitation. We keep the structure of the classical loss and aim at minimizing the error between the backward process $\{\overleftarrow{Y}_t\}_{t=0}^{T}$ and its variational approximation $\{\overleftarrow{Y}_t^{\theta}\}_{t=0}^{T}$. To do so we consider the same loss structure as before, but take the square

root of individual KL terms. See Appendix C.4.3 for a more principled approach leading to a similar loss. Thus we consider the following valid loss function:

$$\mathscr{L}^{\mathrm{L}} : \theta \mapsto \mathscr{L}^{\mathrm{L}}(\theta) = \mathbb{E}\left[\sum_{t=2}^{T} \left(\mathscr{L}_{t-1}^{\mathrm{L}}(\theta, A_{1:t})\right)^{r}\right] , \qquad (19)$$

where $r > 0$,

$$\mathscr{L}_{t-1}^{\mathrm{L}}(\theta, A_{1:t}) = \mathbb{E}\left[\mathrm{KL}\left(k_{t-1|t,0,a}^{(\alpha)}(\cdot|Y_t, Y_0, A_{1:t}) \,\|\, \overleftarrow{q}_{t-1|t,a}^{\theta}(\cdot|Y_t, A_{1:t})\right) \,\Big|\, A_{1:t}\right] ,$$

and $k_{t-1|0,t,a}^{(\alpha)}$ denotes the conditional density of $Y_{t-1}$ given $Y_0, Y_t$ and $\{A_t\}_{t=1}^{T}$. In order to ensure that the expectations with respect to $A_{1:T}$ are finite, we need to choose $r < \frac{\alpha}{2}$ when $\alpha \in (1, 2)$. For simplicity, in the rest of the paper, we will use $r = \frac{1}{2}$.

**Proposition 7** (Training loss for DLPM). *The loss $\mathscr{L}^{\mathrm{L}}$ admits a closed-form expression, such that one resorts to optimize the following loss for $1 \leqslant t \leqslant T$:*

$$\mathscr{L}_{t-1}^{\mathrm{L}}(\theta, A_{1:t}) = \mathbb{E}\left[\frac{1}{2}\log\frac{\hat{\Sigma}_{t-1}^{\theta}}{\tilde{\Sigma}_{t-1}} + \frac{\tilde{\Sigma}_{t-1} + \|\tilde{\mathrm{m}}_{t-1} - \hat{\mathrm{m}}_{t-1}^{\theta}\|^2}{2\hat{\Sigma}_{t-1}^{\theta}} - \frac{1}{2} \,\Big|\, A_{1:t}\right]$$

*where*

$$\tilde{\mathrm{m}}_{t-1}(Y_t, Y_0, A_{1:t}) = \frac{1}{\gamma_t}\left(Y_t - \sigma_{1\to t}\Gamma_t(A_{1:t})\epsilon_t(Y_t, Y_0)\right)$$

$$\tilde{\Sigma}_{t-1}(A_{1:t}) = \Gamma_t(A_{1:t})\Sigma_{1\to t-1}(A_{1:t-1})$$

$$\epsilon_t(Y_t, Y_0) = \frac{Y_t - \gamma_{1\to t}Y_0}{\sigma_{1\to t}}$$

$$\Sigma_{1\to t}(A_{1:t}) = \sum_{k=1}^{t}\left(\frac{\gamma_{1\to t}}{\gamma_{1\to k}}\sqrt{A_k}\sigma_k\right)^2$$

$$\Gamma_t(A_{1:t}) = 1 - \frac{\gamma_t^2\Sigma_{1\to t-1}(A_{1:t-1})}{\Sigma_{1\to t}(A_{1:t})} ,$$

*where $\hat{\mathrm{m}}_{t-1}^{\theta}, \hat{\Sigma}_{t-1}^{\theta}$ are the mean and variance of the backward transition kernels $\overleftarrow{q}_{t-1|t}^{\theta}$. We have omitted the arguments of the mean and variance functions for clarity.*

*Proof.* Recall (Proposition 6) that the backward process $Y_{t-1}$ conditioned on $\{A_t\}_{t=1}^{T}, Y_t, Y_0$ at time $t$ is distributed as $\mathcal{N}(\tilde{\mathrm{m}}_{t-1}, \tilde{\Sigma}_{t-1})$, and, by design (Section 3.2.2), the backward transition kernels $\overleftarrow{q}_{t-1|t,a}^{\theta}$ of each element of the variational family describe a Gaussian transition kernel of mean $\hat{\mathrm{m}}_{t-1}^{\theta}$ and variance $\hat{\Sigma}_{t-1}^{\theta}$ at time $t$, as defined in 14. Since the KL term in $\mathscr{L}_{t-1}^{\mathrm{L}}(\theta, A_{1:t})$ corresponds to a KL divergence between two Gaussian distributions, a closed-form formula is readily available. Here we rewrite the equation with all the functions arguments written out explicitly:

$$\mathscr{L}_{t-1}^{\mathrm{L}}(\theta, A_{1:t}) = \mathbb{E}\left[\frac{1}{2}\log\frac{\hat{\Sigma}_{t-1}^{\theta}(A_{1:t})}{\tilde{\Sigma}_{t-1}(A_{1:t})} - \frac{1}{2}\right. \qquad (20)$$

$$\left. + \frac{\tilde{\Sigma}_{t-1}(A_{1:t}) + \|\tilde{\mathrm{m}}_{t-1}(Y_t, Y_0, A_{1:t}) - \hat{\mathrm{m}}_{t-1}^{\theta}(Y_t, A_{1:t})\|^2}{2\hat{\Sigma}_{t-1}^{\theta}(A_{1:t})} \,\Big|\, A_{1:t}\right]$$

$\square$

Now we discuss further design choices for $\mathscr{L}_{t-1}^{\mathrm{L}}(\theta, A_{1:t}), \overleftarrow{q}_{t-1|t,a}^{\theta}$, leading to a simplified denoising loss, as is usually done in the literature. We denote them by **D1, D2** and **D3**.

**D1.** We set a fixed variance $\hat{\Sigma}_t^{\theta} = \tilde{\Sigma}_t$ for the reverse process , but we expect a study on the effect of learning variance to yield similar results as in original DDPM (Ho et al. (2020)), and especially its improved version (Nichol & Dhariwal (2021)).

**D2.** Following our own experimental results and the usual recommendation for denoising diffusion models (see, e.g., Yang et al. (2024); Karras et al. (2022)), we reparameterize the output of the model to predict the value $\epsilon_t(y_t, y_0)$ at time-step $t$ rather than $\tilde{\mathrm{m}}_{t-1}(y_t, y_0, a_{1:t})$. Since

$$\tilde{\mathrm{m}}_{t-1}(Y_t, Y_0, A_{1:t}) = \frac{1}{\gamma_t} \left( Y_t - \sigma_{1\to t} \Gamma_t(A_{1:t}) \epsilon_t(Y_t, Y_0) \right) ,$$

we re-parameterize $\hat{\mathrm{m}}_{t-1}^\theta$ to be equal to

$$\hat{\mathrm{m}}_{t-1}^\theta(Y_t, A_{1:t}) = \frac{1}{\gamma_t} \left( Y_t - \sigma_{1\to t} \Gamma_t(A_{1:t}) \hat{\epsilon}_t^\theta(Y_t, A_{1:t}) \right) .$$

with $\hat{\epsilon}_t^\theta$ being the output of the model. Following experimental results (see the introduction of Appendix G), we drop the dependency of $\hat{\epsilon}_t^\theta$ on $\{A_t\}_{t=1}^T$, which is reasonable since it approximates $\epsilon_t : (Y_t, Y_0) \mapsto \epsilon_t(Y_t, Y_0)$. Thus $\hat{\epsilon}_t^\theta$ only depends on $t, Y_t$. This choice achieves better performance in our experiments, allows further computational tricks (introduced in Appendix C.5), and enables one to re-use existing neural network architectures.

**D3.** Assuming D1, D2, $\mathscr{L}_{t-1}^{\mathrm{L}}(\theta)$ becomes

$$\mathscr{L}_{t-1}^{\mathrm{L}}(\theta) = \mathbb{E}\left[ \lambda_{t,A_{1:t}}^2 \| \hat{\epsilon}_t^\theta(Y_t, A_{1:t}) - \epsilon_t(Y_t, Y_0) \|^2 \right] , \tag{21}$$

where $\lambda_{t,a_{1:t}} = \Gamma_t(a_{1:t}) \sigma_{1\to t} / 2\gamma_t \tilde{\Sigma}_{t-1}$, and $\epsilon_t(Y_t, Y_0) = (Y_t - \gamma_{1\to t} Y_0)/\sigma_{1\to t}$. The methodological knowledge of diffusion models motivates making specific choices for $\lambda$, different from its defined value, resulting in a classical technique for improving performances (e.g., see Karras et al. (2022); Ho et al. (2020); Nichol & Dhariwal (2021); Yang et al. (2024)). We will stick to the classical choice of choosing $\lambda_{t,a_{1:t}} = 1$, which experimentally works better and draws similarities to the score-based perspective (see Appendix E). Other choices and optimizations are left to further work.

The proof of Proposition 2 follows immediately from these design choices, hence omitted.

### C.4.3 A PRINCIPLED APPROACH FOR DERIVING THE LOSS FUNCTION

In this section, we provide a more principled approach to derive the loss function for DLPM, as initially given in (19). We show the derivation for $r = 1/2$; however, the same derivation applies for any $r \in (0, 1]$.

Noting that $Y_0$ is independent of $\{A_t\}_{t=1}^T$ in (5), $p_\star$ is the equal to $k_{0|a}^{(\alpha)}(\cdot|a_{1:T})$, the conditional density of $Y_0$ given $A_t = a_t$ for any $t \in \{1, \ldots, T\}$, and therefore, we consider the valid loss function

$$\mathscr{L}^{\mathrm{L}}(\theta) : \theta \mapsto \int \mathrm{d}a_{1:T} \psi_{1:T}^{(\alpha)}(a_{1:T}) \left[ \mathrm{KL}(k_{0|a}^{(\alpha)}(\cdot|a_{1:T}) \| \overleftarrow{q}_{0|a}^\theta(\cdot|a_{1:T})) \right]^{1/2} .$$

While this function is still intractable, we can provide an upper bound which we can minimize. Indeed, using Jensen inequality twice, we bound this function by

$$\mathscr{L}^{\mathrm{L}}(\theta) = \int \mathrm{d}a_{1:T} \psi_{1:T}^{(\alpha)}(a_{1:T}) \left\{ \int \mathrm{d}y_0 k_{0|a}^{(\alpha)}(y_0|a_{1:T})(\log k_{0|a}^{(\alpha)}(y_0|a_{1:T}) - \log \overleftarrow{q}_{0|a}^\theta(y_0|a_{1:T})) \right\}^{1/2}$$

$$\leqslant \int \mathrm{d}a_{1:T} \psi_{1:T}^{(\alpha)}(a_{1:T}) \left\{ -\int \mathrm{d}y_{0:T} k_{0:T|0,a}^{(\alpha)}(y_{0:T}|a_{1:T}) \log \frac{\overleftarrow{q}_{0:T|a}^\theta(y_{0:T}|a_{1:T})}{k_{1:T|0,a}^{(\alpha)}(y_{1:T}|y_0,a_{1:T})} + \mathrm{Cst}_1 \right\}^{1/2}$$

$$= \int \mathrm{d}a_{1:T} \psi_{1:T}^{(\alpha)}(a_{1:T}) \{ \textstyle\sum_{t=0}^{T-1} \mathscr{L}_t^{\mathrm{L}}(\theta, a_{1:T}) + \mathrm{Cst}_1 + \mathrm{Cst}_2 \}^{1/2}$$

$$\leqslant \int \mathrm{d}a_{1:T} \psi_{1:T}^{(\alpha)}(a_{1:T}) \{ \textstyle\sum_{t=0}^{T-1} \mathscr{L}_t^{\mathrm{L}}(\theta, a_{1:T})^{1/2} + \mathrm{Cst}_1^{1/2} + \mathrm{Cst}_2^{1/2} \} ,$$

where we used $\sqrt{a+b} < \sqrt{a} + \sqrt{b}$ when $a, b \geqslant 0$ and $\mathscr{L}_0^{\mathrm{L}}(\theta, a_{1:T}) = \mathbb{E}[-\log p_\theta(Y_0|Y_1, a_{1:T})|\{A_t\}_{t=1}^T = \{a_t\}_{t=1}^T]$ for $t > 0$,

$$\mathscr{L}_t^{\mathrm{L}}(\theta, a_{1:T}) = \mathbb{E}\left[ \mathrm{KL}(k_{t|t+1,0,a}^{(\alpha)}(\cdot|Y_t, Y_0, a_{1:T}) \| \overleftarrow{q}_{t|t+1,a}^\theta(\cdot|Y_t, a_{1:T})) | \{A_t\}_{t=1}^T = \{a_t\}_{t=1}^T \right] ,$$

and $\mathrm{Cst}_1 = \int \mathrm{d}y_0 p_\star(y_0) \log p_\star(y_0) \mathrm{d}y_0$ and $\mathrm{Cst}_2 = \mathbb{E}[\mathrm{KL}(k_{1:T|0,a}^{(\alpha)}(\cdot|Y_0, a_{1:T}) \| \overleftarrow{q}_T^\theta) | \{A_t\}_{t=1}^T = \{a_t\}_{t=1}^T]$ does not depend on $\theta$ since $\overleftarrow{q}_T^\theta$ is chosen as $\mathcal{S}_\alpha(0, \sigma_{1\to T} \mathrm{I}_d)$. Regarding $\mathscr{L}_0^{\mathrm{L}}$, we neglect this

term, replacing the distribution $\overleftarrow{q}^{\theta}_{0|1,a}(\cdot|y_1, a_{1:T})$ by a deterministic Dirac. One could alternatively employ the strategy of the discrete decoder for image data as described by Ho et al. (2020). We end up with the final loss function:

$$\mathscr{L}^{\mathrm{L}}(\theta) = \int \mathrm{d}a_{1:T} \psi^{(\alpha)}_{1:T}(a_{1:T}) \{ \textstyle\sum_{t=0}^{T-1} \mathscr{L}^{\mathrm{L}}_t(\theta, a_{1:T})^{1/2} \} \ .$$

We can then provide an explicit expression for $\mathscr{L}^{\mathrm{L}}_t(\theta, a_{1:T})$ based on the result of Proposition 6.

## C.5 Reducing the computational cost with faster sampling at each timestep

In this section, we provide a faster algorithm for training DLPM, computing each loss term $\mathscr{L}^{\mathrm{L}}_{t-1}$ using only two heavy-tailed random variables per datapoint, instead of $t$ random variables.

Consider again the process $\{Y_t\}_{t=0}^T$. We replace the loss function (19) by an equivalent one:

$$\mathscr{L}^{\mathrm{L}}_{\mathrm{time}}(\theta) = \mathbb{E}\left[ \mathscr{L}^{\mathrm{L}}_{t-1}(\theta, A_{1:t})^{1/2} \right] \ , \tag{22}$$

where $t \sim \mathcal{U}[2, T] = (\sum_{i=2}^T \delta_i)/(T-1)$. The standard technique for computing the loss consists in the following loop:

1. Take a batch of $B$ datapoints $\{Y_0^i\}_{i=1}^B$.
2. For each datapoint $Y_0^i$, draw a random $t_i$, as suggested by the alternative loss (22). Indeed, for a single datapoint, (i) training on all timesteps rather than just one yields equal to inferior results, for a much higher computational cost, (ii) it is beneficial for the model to proportionally spend more time learning specific time ranges (iii) thus the distribution of $t$ can be optimized and is a matter of ongoing methodological research, e.g., see Karras et al. (2022).
3. Draw sequences $\{A_t^i\}_{t=1}^{t_i}$ of heavy-tailed random variables.
4. Compute the noised datapoints $\{Y_{t_i}^i\}_{i=1}^B$.
5. Compute the batch loss

$$\hat{\mathscr{L}}^{\mathrm{L}}(\theta) = \frac{1}{B} \sum_{i=1}^B \mathscr{L}^{\mathrm{L}}_{t_i-1}(\theta, Y_{t_i}^i, A_{1:t_i}^i) \ ,$$

such that $\hat{\mathscr{L}}^{\mathrm{L}}(\theta) \approx \mathscr{L}^{\mathrm{L}}_{\mathrm{time}}(\theta)$.
6. Do an optimization step.

Step 3 can be expensive, since one has to sample on average $O(T)$ $d$-dimensional heavy-tailed random variables to compute a single noised datapoint $Y_t$ from $Y_0$. This is all the more inefficient as $T$ can be quite large (indeed, on image datasets we can have $T = 4000$, see Appendix G).

One can guess that this is abusive, especially since characterizing the distribution of $Y_t$ given $Y_0$ only requires a single heavy-tailed random variable:

$$Y_t \overset{d}{=} \gamma_{1 \to t} Y_0 + \sigma_{1 \to t} \bar{A}^{1/2} \bar{G}_t \ ,$$

where $\bar{A} \sim \mathcal{S}_{\alpha/2,1}(0, c_A)$, $\bar{G}_t \sim \mathcal{N}(0, \mathrm{I}_d)$. As we formalize in the next proposition, it is indeed possible to bypass the sampling of a whole sequence. Since we manipulate the joint distribution of $(Y_0, Y_{t-1}, Y_t)$ for the loss term $\mathscr{L}^{\mathrm{L}}_{t-1}(\theta)$, we will actually need to sample two heavy-tailed random variables.

**Proposition 8** (Sampling two heavy-tailed r.v for each loss term)**.** *Suppose that the functions $\hat{\mathrm{m}}^{\theta}_{t-1}, \hat{\Sigma}^{\theta}_{t-1}$ satisfy for any $y_t, a_{1:t}$:*

$$\hat{\mathrm{m}}^{\theta}_{t-1}(y_t, a_{1:t}) = M^{\theta}_{t-1}\left( y_t, a_t, \frac{\Sigma_{1 \to t-1}(a_{1:t-1})}{\sigma^2_{1 \to t-1}} \right)$$

$$\hat{\Sigma}^{\theta}_{t-1} = S^{\theta}_{t-1}\left( y_t, a_t, \frac{\Sigma_{1 \to t-1}(a_{1:t-1})}{\sigma^2_{1 \to t-1}} \right) \ ,$$

*for some functions $M_{t-1}^\theta, S_{t-1}^\theta$, and where*

$$\Sigma_{1\to t}(a_{1:t}) = \sum_{k=1}^{t} \left( \frac{\gamma_{1\to t}}{\gamma_{1\to k}} \sqrt{a_k}\sigma_k \right)^2 ,$$

*as given in* (12). *Then each term* $\mathbb{E}[\mathscr{L}_{t-1}^L(\theta, A_{1:t})]^{1/2}$ *of the loss can be computed sampling only two independent random variables* $\bar{A}_0^t, \bar{A}_1^t$ *distributed as* $\mathcal{S}_{\alpha/2,1}(0, c_A)$:

$$\mathbb{E}\left[\mathscr{L}_{t-1}^L(\theta, A_{1:t})\right]^{1/2} = \mathbb{E}\left[\mathscr{L}_{t-1}^{\text{Less}}(\theta, \bar{A}_{0,1}^t)\right]^{1/2} ,$$

*where* $\bar{A}_{0,1}^t := (\bar{A}_0^t, \bar{A}_1^t)$, *and*

$$\mathscr{L}_{t-1}^{\text{Less}}(\theta, \bar{A}_{0,1}^t) = \mathbb{E}\left[ \frac{1}{2}\log\frac{S_{t-1}^\theta(Z_t, \bar{A}_{0,1}^t)}{\tilde{\Sigma}_{t-1}(\bar{A}_{0,1}^t)} - \frac{1}{2} \right. \tag{23}$$
$$\left. + \frac{\tilde{\Sigma}_{t-1}(\bar{A}_{0,1}^t) + \|\tilde{\mathrm{m}}_{t-1}'(Z_t, Z_0, \bar{A}_{0,1}^t) - M_{t-1}^\theta(Z_t, \bar{A}_{0,1}^t)\|^2}{2S_{t-1}^\theta(Z_t, \bar{A}_{0,1}^t)} \;\middle|\; \bar{A}_{0,1}^t \right] ,$$

*with* $\{Z_t\}_{t=0}^T$ *being a stochastic process defined as*

$$Z_0 = Y_0 , \quad Z_t = \gamma_{1\to t}Z_0 + \Sigma_t'^{1/2}\bar{G}_t ,$$

*where* $\{G_t\}_{t=1}^T$ *is an i.i.d. sequence distributed as* $\mathcal{N}(0, \mathrm{I}_d)$, *and*

$$\Sigma_{t-1}'(\bar{A}_0^t) = \sigma_{1\to t-1}^2\bar{A}_0^t$$
$$\Sigma_t'(\bar{A}_{0,1}^t) = \sigma_t^2\bar{A}_1^t + \gamma_t^2\Sigma_{t-1}'(\bar{A}_0^t)$$
$$\Gamma_t'(\bar{A}_{0,1}^t) = 1 - \frac{\gamma_t^2\Sigma_{t-1}'(\bar{A}_0^t)}{\Sigma_t'(\bar{A}_{0,1}^t)} ,$$

*such that* $Z_t \overset{d}{=} Y_t$, *and:*

$$\tilde{\mathrm{m}}_{t-1}(Z_t, Z_0, \bar{A}_{0,1}^t) = \frac{1}{\gamma_t}\left( Z_t - \sigma_{1\to t}\Gamma_t'(\bar{A}_{0,1}^t)\epsilon_t(Z_t, Z_0) \right)$$
$$\tilde{\Sigma}_{t-1}(\bar{A}_{0,1}^t) = \Gamma_t'(\bar{A}_{0,1}^t)\Sigma_{t-1}'(\bar{A}_0^t)$$
$$\epsilon_t(Z_t, Z_0) = \frac{Z_t - \gamma_{1\to t}Z_0}{\sigma_{1\to t}} .$$

*In order to keep the notations similar for all* $t \geqslant 2$, *in the case of* $\mathscr{L}_1^{\text{Less}}$, *we always set* $\bar{A}_0^2 = 0$.

*Proof.* Remember the full equation for the loss, first given in Proposition 7 (20):

$$\mathscr{L}_{t-1}^L(\theta, A_{1:t}) = \mathbb{E}\left[ \frac{1}{2}\log\frac{\hat{\Sigma}_{t-1}^\theta(A_{1:t})}{\tilde{\Sigma}_{t-1}(A_{1:t})} - \frac{1}{2} \right.$$
$$\left. + \frac{\tilde{\Sigma}_{t-1}(A_{1:T}) + \|\tilde{\mathrm{m}}_{t-1}(Y_t, Y_0, A_{1:T}) - \hat{\mathrm{m}}_{t-1}^\theta(Y_t, A_{1:T})\|^2}{2\hat{\Sigma}_{t-1}^\theta(A_{1:t})} \;\middle|\; A_{1:t} \right] .$$

Now, all the required variables and computations only depend on $A_t, \Sigma_{1\to t-1}$; this is the case for $\hat{\mathrm{m}}_{t-1}^\theta, \hat{\Sigma}_{t-1}^\theta$ by hypothesis, and this is the case for $\tilde{\mathrm{m}}_{t-1}, \tilde{\Sigma}_{t-1}$ as one can see in (13). Rewriting the previous loss as

$$\mathscr{L}_{t-1}^L(\theta, A_{1:t}) = \mathbb{E}\left[ \frac{1}{2}\log\frac{S_{t-1}^\theta\left(Z_t, A_t, \frac{\Sigma_{1\to t-1}(A_{1:t-1})}{\sigma_{1\to t-1}^2}\right)}{\tilde{\Sigma}_{t-1}(A_{1:T})} - \frac{1}{2} \right.$$
$$+ \frac{\tilde{\Sigma}_{t-1}(A_{1:t})}{2S_{t-1}^\theta\left(Z_t, A_t, \frac{\Sigma_{1\to t-1}(A_{1:t-1})}{\sigma_{1\to t-1}^2}\right)}$$
$$\left. + \frac{\|\tilde{\mathrm{m}}_{t-1}(Y_t, Y_0, A_{1:t}) - M_{t-1}^\theta\left(Z_t, A_t, \frac{\Sigma_{1\to t-1}(A_{1:t-1})}{\sigma_{1\to t-1}^2}\right)\|^2}{2S_{t-1}^\theta\left(Z_t, A_t, \frac{\Sigma_{1\to t-1}(A_{1:t-1})}{\sigma_{1\to t-1}^2}\right)} \;\middle|\; A_t, \Sigma_{1\to t-1} \right] ,$$

it becomes clear how the expectation can be taken on the joint distribution of

$$\left( Y_0, Y_{t-1}, Y_t, A_t, \frac{\Sigma_{1 \to t-1}(A_{1:t-1})}{\sigma_{1 \to t-1}^2} \right) \ .$$

A direct application of Lemma 1 shows that this expectation can be taken on the joint distribution of the five random variables $(Z_0, Z_{t-1}, Z_t, \bar{A}_1^t, \bar{A}_0^t)$, which only necessitates sampling two heavy-tailed random variables $\bar{A}_0^t, \bar{A}_1^t$. Using the formulas for $Z_0, Z_{t-1}$ and $Z_t$ given $\bar{A}_0^t, \bar{A}_1^t$ as defined in Lemma 1, we obtain the equivalent loss (23).

$\square$

As we will prove in the next proposition, the conditions of Proposition 8 are always satisfied under design choices D1, D2. Under design choice D3, we can also rewrite the simplified denoising loss given in Proposition 2.

**Proposition 9** (Sampling one heavy-tailed r.v in the simplified loss)**.** *Assume the design choices D1, D2, D3 are satisfied. Then one can obtain the following simplified denoising objective function from the full objective function given in* (23)*:*

$$\mathscr{L}^{\mathrm{SimpleLess}}(\theta) = \sum_{t=1}^{T} \mathbb{E} \left[ \mathbb{E} \left( \|\hat{\epsilon}_t^\theta(Z_t) - \bar{A}_t^{1/2} \bar{G}_t \|^2 \, |\bar{A}_t \right)^{1/2} \right] ,$$

*where $\{\bar{A}_t\}_{t=1}^{T}$ is an i.i.d. sequence distributed as $\mathcal{S}_{\alpha/2,1}(0, c_A)$, and*

$$Z_t = \gamma_{1 \to t} Z_0 + \sigma_{1 \to t} \bar{A}_t^{1/2} \bar{G}_t \ ,$$

*with $\{\bar{G}_t\}_{t=1}^{T}$ an i.i.d. sequence distributed as $\mathcal{N}(0, \mathrm{I}_d)$.*

*Proof.* Let us recall design choice D1:

$$\hat{\Sigma}_{t-1}^\theta(A_{1:}) = \Gamma_t(A_{1:t}) \Sigma_{1 \to t}(A_{1:t}) \ ,$$

and design choice D2:

$$\hat{\mathrm{m}}_{t-1}^\theta(Z_t, A_{1:t}) = \frac{1}{\gamma_t} \left( Z_t - \sigma_{1 \to t} \Gamma_t(A_{1:t}) \hat{\epsilon}_t^\theta(Z_t) \right) \ .$$

Since $\Gamma$ only depends on $\Sigma_{1 \to t}$ and $\Sigma_{1 \to t-1}$, both $\hat{\mathrm{m}}_{t-1}^\theta, \tilde{\Sigma}_{t-1}$ can be expressed as functions of $z_t, a_t$ and $\Sigma_{1 \to t}(a_{1:t})$. Thus the assumptions of Proposition 8 are satisfied. Using the same notations, we can apply the same algebraic transformations as in (21), and by design choice D3, obtain:

$$\mathscr{L}^{\mathrm{Less}}(\theta) = \sum_{t=1}^{T} \mathbb{E} \left[ \mathbb{E} \left( \|\hat{\epsilon}_t^\theta(Z_t) - \epsilon_t(Z_t, Z_0)\|^2 \, |\bar{A}_0^t, \bar{A}_1^t \right)^{1/2} \right] \ .$$

Finally, we apply Lemma 2 to $\Sigma_t'$ to affirm that $\Sigma_t' \stackrel{d}{=} \sigma_{1 \to t}^2 \bar{A}_t$, where $\bar{A}_t \sim \mathcal{S}_{\alpha/2,1}(0, c_A)$, and obtain the final loss we presented. $\square$

We stress that this denoising training loss is similar to that of LIM (Yoon et al., 2023, Theorem 4.3), but elevated to the necessary power to guarantee that the loss is finite. See Appendix E.1 for a more detailed discussion.

## D  DENOISING LÉVY IMPLICIT MODELS (DLIM)

Using the same techniques as in DDIM (Song et al. (2020)), we obtain a deterministic sampling process which we naturally call Denoising Levy Implicit Models (DLIM). Alike the Gaussian case treated in the original DDIM work, we will show that both DLPM and DLIM can share the same neural network.

### D.1 Non-Markovian Forward Process

Let $\{\rho_t\}_{t=1}^T$ be an alternative noise schedule, proper to DLIM, that will ultimately tend to zero for deterministic generation. In the same way as in Section 3.2.2, we take a data augmentation approach. We consider a process $\{Z_t\}_{t=1}^T$ defined by $Z_0 \sim p_\star$, where $p_\star$ is the data distribution, $Z_T \sim \mathcal{S}(\gamma_{1\to T} Z_0, \sigma_{1\to T} \mathrm{I}_d)$ and, for $1 < t \leqslant T$

$$Z_{t-1} = \gamma_{1\to t-1} Z_0 + (\sigma_{1\to t-1}^\alpha - \rho_t^\alpha)^{1/\alpha} \epsilon_t(Z_t, Z_0) + \rho_t A_t^{1/2} G_t$$

where $\epsilon_t(Z_t, Z_0) = (Z_t - \gamma_{1\to t} Z_0)/\sigma_{1\to t}$, and $\{A_t\}_{t=1}^T$, $\{G_t\}_{t=1}^T$ are independent random variables distributed according to $A_t \sim \mathcal{S}_{\alpha/2,1}(0, c_A)$ and $G_t \sim \mathcal{N}(0, \mathrm{I}_d)$.

**Proposition 10.** *The distribution of $Z_t$ given $Z_0$ is the same as that of $Y_t$ given $Y_0$.*

*Proof.* This is a simple proof by induction, where one can re-adapt the technique of (Song et al., 2020, Lemma B.1) with the property for addition of $\alpha$-stable variable as we introduced in Proposition 3. The case $t = T$ is true by construction. Suppose now that the property is verified at timestep $t$, where $1 \leqslant t \leqslant T$. Then, focusing on the distribution of $Z_{t-1}$ given $Z_0$, $\epsilon_t(Z_t, Z_0) = (Z_t - \gamma_{1\to t} Z_0)/\sigma_{1\to t}$ is distributed as $\mathcal{S}_\alpha^i(0, \mathrm{I}_d)$ by hypothesis, and thus by Proposition 3 and since $A_t^{1/2} G_t \sim \mathcal{S}_\alpha^i(0, \mathrm{I}_d)$, we can write

$$Z_{t-1} \stackrel{d}{=} \gamma_{1\to t-1} Z_0 + \sigma_{1\to t-1} \bar\epsilon_t ,$$

where $\bar\epsilon_t \sim \mathcal{S}_\alpha^i(0, \mathrm{I}_d)$, which shows that indeed $Z_{t-1}$ given $Z_0$ admits the same distribution as $Y_{t-1}$ given $Y_0$. $\qquad\square$

The design of this process makes the distribution of $Z_t$ given $Z_0$ match that of $Y_t$ given $Y_0$, where $\{Y_t\}_{t=0}^T$ is the forward process of DLPM (5). The task of sampling from it is thus efficient and straightforward.

### D.2 Generative Process

We similarly reconsider the family of Gaussian variational approximation introduced in (4), which accounts for an i.i.d. sequence $\{A_t\}_{t=1}^T$:

$$\overleftarrow{q}_{0:T}^\theta(y_{0:T}) = \int \psi_{1:T}^{(\alpha)}(a_{1:T}) p_T(y_T) \prod_{t=1}^T \overleftarrow{q}_{t-1|t,a}^\theta(y_{t-1}|y_t, a_{1:t}) \mathrm{d}a_{1:T} ,$$

where $\psi_{1:T}^{(\alpha)}(a_{1:T}) = \prod_{i=1}^T \psi^{(\alpha)}(a_i)$, $\psi^{(\alpha)}$ is the density of $\mathcal{S}_{\alpha/2,1}(0, c_A)$, and $p_T$ is the density of $\mathcal{S}(0, \sigma_{1\to T} \mathrm{I}_d)$. We set

$$\overleftarrow{q}_{t-1|t,a}^\theta(z_{t-1}|z_t, a_{1:t}) = \phi_d\left(z_{t-1}|\hat{\mathrm{m}}_{t-1}^\theta(z_t), \rho_t^2 a_t\right) ,$$

with $\phi_d$ being the density of the multivariate Gaussian distribution: the variance is fixed, determined by the alternative noise schedule $\{\rho_t\}_{t=1}^T$. For deterministic sampling, one will ultimately choose $\rho_t = 0$ for all $t$ and sample the chain $\{\overleftarrow{Z}_t^\theta\}_{t=0}^T$ as follows:

$$\overleftarrow{Z}_T^\theta \sim \mathcal{S}(0, \sigma_{1\to T} \mathrm{I}_d) , \quad \overleftarrow{Z}_{t-1}^\theta = \hat{\mathrm{m}}_t^\theta(\overleftarrow{Z}_t^\theta) \text{ for } t \in \{T, \cdots, 1\} .$$

As $z_t$ is available as input, the model can be fit to approximate the value of $\epsilon_t(z_t, z_0)$, and we reparameterize $\hat{\mathrm{m}}_t^\theta$ as follows:

$$\hat{\mathrm{m}}_t^\theta(z_t) = \frac{z_t - \sigma_{1\to t} \hat\epsilon_t^\theta(z_t)}{\gamma_t} + (\sigma_{1\to t-1}^\alpha - \rho_t^\alpha)^{1/\alpha} \hat\epsilon_t^\theta(z_t) . \tag{24}$$

This is alike design choice D2 in Appendix C.4.2.

### D.3 Loss function and equivalence with DLPM

We denote by $h^{(\alpha)}_{t-1|t,0,a}$ the density of $Z_{t-1}$ given $Z_t$, $Z_0$ and $A_{1:T}$, which is the density of the Gaussian distribution with mean $\gamma_{1\to t-1}Z_0 + (\sigma^\alpha_{1\to t-1} - \rho^\alpha_t)^{1/\alpha}\epsilon_t(Z_t, Z_0)$ and covariance $\rho^2_t A_t \mathrm{I}_d$. Since this distribution is now a given, we are inclined to use the loss function introduced in (7), which is:

$$\mathscr{L}^{\mathrm{L}}(\theta) := \mathbb{E}\left[\sum_{t=2}^T \left(\mathscr{L}^{\mathrm{L}}_{t-1}(\theta, A_{1:t})\right)^{1/2}\right], \qquad \text{where}$$

$$\mathscr{L}^{\mathrm{L}}_{t-1}(\theta, A_{1:t}) := \mathbb{E}\left[\mathrm{KL}\left(h^{(\alpha)}_{t-1|t,0,a_{1:t}}(\cdot|Z_t, Z_0, A_{1:t}) \,\|\, \overleftarrow{q}^{\,\theta}_{t-1|t,a}(\cdot|Z_t, A_{1:t})\right) \Big| A_{1:T}\right] .$$

Since for $2 \leqslant t \leqslant T$, $h^{(\alpha)}_{t-1|t,0,a}$ and $\overleftarrow{q}^{\,\theta}_{t-1|t,a}$ are the densities of Gaussian distributions, we can analytically compute each term of the loss, as in (19):

$$\mathscr{L}^{\mathrm{L}}_{t-1}(\theta, A_{1:t}) = \frac{1}{2\rho^2_t A_t}\|\gamma_{1\to t-1}Z_0 + (\sigma^\alpha_{1\to t-1} - \rho^\alpha_t)^{1/\alpha}\epsilon_t(Z_t, Z_0) \,-\, \hat{\mathtt{m}}^\theta_t(Z_t, A_{1:t})\|^2$$

where $\epsilon_t(Z_t, Z_0) = (Z_t - \gamma_{1\to t}Z_0)/\sigma_{1\to t}$. Since the variance of the elements of our variational family $\{\overleftarrow{q}^{\,\theta}_{0:T}\}$ have been designed to match that of the backward process given $Z_0, A_{1:T}$, the expression for the loss is readily in a simpler format. Finally, using the reparameterization given in (24), The loss term $\mathscr{L}^{\mathrm{L}}_{t-1}(\theta, A_{1:T})$ becomes:

$$\mathscr{L}^{\mathrm{L}}_{t-1}(\theta, A_{1:t}) = \lambda'_{t,A_t}\|\epsilon_t(Z_t, Z_0) - \hat{\epsilon}^\theta_t(Z_t)\|^2 ,$$

where $\lambda'_{t,a_t} = (\sigma_{1\to t} - (\sigma^\alpha_{1\to t-1} - \rho^\alpha_t)^{1/\alpha})^2/(2\rho^2_t a_t)$. By comparing with the simpler DLPM loss (21) with design choices D1, D2, as introduced in Appendix C.4.2, we realize we obtained the same loss term, with a different multiplicative factor $\lambda'_{t,a_t}$ instead of $\lambda_{t,a_{1:t}}$ in (21). Finally, considering the alternative loss where $\lambda'_{t,a_t} = 1$ for all $t$, alike the design choice D3 in Appendix C.4.2, we fall back to the same simplified objective function obtained for DLPM:

$$\mathscr{L}^{\mathrm{Simple}}(\theta) = \sum_{t=1}^T \mathbb{E}\left[\mathbb{E}\left(\|\hat{\epsilon}^\theta_t(Y_t) - \epsilon_t(Y_t, Y_0)\|^2 \,\big|\, A_{1:t}\right)^{1/2}\right] ,$$

### D.4 Cauchy DLIM

In the special case of a non-isotropic Cauchy distribution ($\alpha = 1$), it is possible to bypass the data augmentation machinery, since there exists a closed-form formula for the KL divergence between two Cauchy distributions. Denote by $\mathrm{Cauchy}(\mu, \sigma)$ the one-dimensional Cauchy distribution centered at $\mu$ and of scale $\sigma$. Then (Chyzak & Nielsen, 2019, Theorem 1):

$$\mathrm{KL}(\mathrm{Cauchy}(\mu_1, \sigma_1) \,\|\, \mathrm{Cauchy}(\mu_2, \sigma_2)) = \log\left(\frac{(\mu_1 - \mu_2)^2 + (\sigma_1 + \sigma_2)^2}{4\sigma_1\sigma_2}\right) . \qquad (25)$$

The forward process $\{Z_t\}_{t=1}^T$ is defined with $Z_0 \sim p_\star$, where $p_\star$ is the data distribution, $Z_T \sim \mathcal{S}(\gamma_{1\to T}Z_0, \sigma_{1\to T}\mathrm{I}_d)$ and, for $1 < t \leqslant T$

$$Z_{t-1} = \gamma_{1\to t-1}Z_0 + (\sigma^\alpha_{1\to t-1} - \rho^\alpha_t)^{1/\alpha}\epsilon_t(Z_t, Z_0) + \rho_t\epsilon^{(\alpha)}_t ,$$

where $\epsilon_t(Z_t, Z_0) = (Z_t - \gamma_{1\to t}Z_0)/\sigma_{1\to t}$, and $\{\epsilon^{(\alpha)}_t\}_{t=1}^T \sim \mathrm{Cauchy}(0, \mathrm{I}_d)^{\otimes T}$, where $\mathrm{Cauchy}(0, \mathrm{I}_d)$ is a $d$-dimensional Cauchy distribution with independent components, centered at 0, of unit scale. The distribution of $Z_t$ given $Z_0$ admits the same closed form expression given in Proposition 10. We denote by $h^{(\alpha)}_{t-1|t,0}$ the density of $Z_{t-1}$ given $Z_t$, $Z_0$.

Our generative process will be an element of a parameterized family of distributions admitting Cauchy transitions:

$$\overleftarrow{q}^{\,\theta}_{0:T}(y_{0:T}) = p_T(y_T)\prod_{t=1}^T \overleftarrow{q}^{\,\theta}_{t-1|t,a}(y_{t-1}|y_t) ,$$

where $p_T$ is the density of $\mathcal{S}(0, \sigma_{1\rightarrow T}\mathrm{I}_d)$, and

$$\overleftarrow{q}^{\theta}_{t-1|t}(z_{t-1}|z_t) = \mathrm{C}\left(z_{t-1}|\hat{\mathrm{m}}^{\theta}_{t-1}(z_t),\ \rho_t\mathrm{I}_d\right)\ ,$$

where $\mathrm{C}(\cdot|\hat{\mathrm{m}}^{\theta}_{t-1}(z_t),\ \rho_t\mathrm{I}_d)$ is the density of the multivariate non-isotropic Cauchy distribution Cauchy$(\hat{\mathrm{m}}^{\theta}_{t-1}(z_t), \rho_t\mathrm{I}_d)$.

Instead of using the loss function $\mathscr{L}^{\mathrm{L}}$ defined in (7), we derive the loss via the conventional evidence lower bound (ELBO) approach (see, e.g., Ho et al. (2020)). Omitting extremal terms, this yields:

$$\mathscr{L}^{\mathrm{Cauchy}}(\theta) := \sum_{t=2}^{T}\mathscr{L}^{\mathrm{Cauchy}}_{t-1}(\theta)\ ,\qquad\text{where}$$

$$\mathscr{L}^{\mathrm{Cauchy}}_{t-1}(\theta) := \mathbb{E}\left[\mathrm{KL}\left(h^{(\alpha)}_{t-1|t,0}(\cdot|Z_t, Z_0)\ \|\ \overleftarrow{q}^{\theta}_{t-1|t}(\cdot|Z_t)\right)\right]\ .$$

Using (25), we obtain a closed form formula for the final loss:

$$\mathscr{L}^{\mathrm{Cauchy}}_{t-1}(\theta) = \sum_{i=1}^{d}\log\left(\frac{\left(\tilde{\mathrm{m}}_{i,t-1}(Z_t, Z_0) - \hat{\mathrm{m}}^{\theta}_{i,t-1}(Z_t)\right)^2}{4\rho_t^2} + 1\right)\ ,$$

which could also serve as a template for another family of losses for heavy-tailed diffusion models. We leave these methodological explorations and possible extensions to the isotropic Cauchy case for future work. Based on our experimental findings, we expect an isotropic implementation to significantly outperform a non-isotropic one.

We outline again that such simplifications are not available for DLPM, since we are not able to characterize the distribution of $X_{t-1}$ given $X_t, X_0$ from the forward process $\{X_t\}_{t=0}^{T}$.

# E  ADDITIONAL INFORMATION ON LEVY-ITO MODELS (LIM)

Here we briefly recapitulate the work done by Yoon et al. (2023), introducing continuous diffusion models with $\alpha$-stable heavy-tailed noise. Using notations closer to ours, we define the noising schedule as any locally bounded continuous functions $\gamma : (t, X) \mapsto \gamma(t, X)$ and $\sigma : (t) \mapsto \sigma(t)$. We denote by $L_t^{\alpha}$ the Levy process for which the increments between time $s < t$ follow a symmetric isotropic $\alpha$-stable distribution $\mathcal{S}^{\mathrm{i}}_{\alpha}(0, (t - s)\mathrm{I}_d)$. In this setting, the forward process $X_t$, with $X_0 \sim p_{\star}$, is written

$$\mathrm{d}X_t = \gamma(t, X_{t-})\mathrm{d}t + \sigma(t)\mathrm{d}L_t^{\alpha}\ , \tag{26}$$

where $X_{t-}$ denotes the left limit of $X$ at time $t$. Similarly, $X_t$ is distributed as $\mathcal{S}^{\mathrm{i}}_{\alpha}(\gamma_{1\rightarrow t}X_0, \sigma_{1\rightarrow t}\mathrm{I}_d)$ when using Euler steps. This defines the cadlag (right continuous with left limits) solution, which in the case of $\alpha < 2$ a.s admits discontinuous jumps. We then consider the following backward process $\overleftarrow{X}_t$:

$$\mathrm{d}\overleftarrow{X}_t = \left(-\gamma(t, \overleftarrow{X}_{t+}) - \alpha\sigma^{\alpha}(t, \overleftarrow{X}_{t+})S_t^{(\alpha)}(\overleftarrow{X}_{t+})\right)\mathrm{d}t + \sigma(t)\mathrm{d}\bar{L}^{\alpha}_t + d\bar{Z}_t \tag{27}$$

where $\bar{Z}_t$ is the backward version of a Levy-type stochastic integral $Z_t$ s.t $\mathbb{E}[Z_t] = 0$ with finite variation, and $S_t^{(\alpha)}$ is the fractional score function, defined to be

$$S_t^{(\alpha)}(x) = \frac{\Delta^{\frac{\alpha-2}{2}}\nabla p_t(x)}{p_t(x)}\ ,$$

where $\Delta^{\eta/2}$ denotes the fractional Laplacian of order $\eta/2$ (Ortigueira et al. (2014)). More precisely, $\Delta^{\eta/2}f(x) = \mathcal{F}^{-1}\{\|u\|^{\eta}\mathcal{F}\{f\}(u)\}$, where $\mathcal{F}, \mathcal{F}^{-1}$ are the Fourier and inverse Fourier transforms.

The training loss is obtained using the classical technique of denoising score matching (Vincent (2011)), where the following losses

$$L : \theta \mapsto \mathbb{E}\|s_{\theta}(X_t, t) - S_t^{(\alpha)}(X_t)\|^2\ ,\qquad L' : \theta \mapsto \mathbb{E}\|s_{\theta}(X_t, t) - S_t^{(\alpha)}(X_t|X_0)\|^2\ , \tag{28}$$

are proven to be equivalent objective functions, with $s_{\theta}$ being the score approximation given by the model.

### E.1 COMPARING LIM AND DLPM

Let $(X_t)_{0 \leqslant t \leqslant T}$ be the forward process introduced in (26). As stressed initially, the framework of LIM is not straightforward to manipulate, thus we do not characterize explicitly the distribution of $X_t$ given $X_0$ for an arbitrary noise schedule in the continuous case. Since the work done for LIM by Yoon et al. (2023) only provides the formulas for the scale-preserving schedule, we stick to them in the following: we keep the notation $\gamma_{1 \to t}, \sigma_{1 \to t}$ for the continuous time regime equivalent of the scale preserving schedule we introduce in Appendix G, and they match on integer times $t$.

Considering an Euler scheme to obtain discretization for the forward and backward process, and using our own notations, both LIM and DLPM admit the same forward process $\{X_t\}_{t=1}^{T}$, $X_0 \sim p_\star$ and

$$X_t = \gamma_t X_{t-1} + \sigma_t \epsilon_t^{(\alpha)},$$

where $\{\epsilon_t^{(\alpha)}\}_{t=1}^{T}$ is an iid sequence of random variable distributed as $\mathcal{S}_\alpha^{\mathrm{i}}(0, \mathrm{I}_d)$. We denote by $\{\overleftarrow{X}_t^\theta\}_{t=T}^{0}$ the backward process associated to the Euler discretization of (27), where we use a neural network $s_\theta$ to approximate the true score $S_t^{(\alpha)}$. Since the true score of the data $S_t^{(\alpha)}(x_t|x_0)$ can be expressed as

$$S_t^{(\alpha)}(x_t|x_0) = -\frac{1}{\alpha \sigma_{1 \to t}^{\alpha-1}} \epsilon_t(x_t, x_0) \ ,$$

where $\epsilon_t(x_t, x_0) = (x_t - \gamma_{1 \to t} x_0)/\sigma_{1 \to t}$, we write

$$s_\theta(x_t, t) = -\frac{1}{\alpha \sigma_{1 \to t}^{\alpha-1}} \hat{\epsilon}_t^\theta(x_t, x_0) \ ,$$

so that we rather work with $\hat{\epsilon}_t^\theta$, with the same intention that led us to the design choices given in Appendix C.4.2.

Moreover, we denote by $\{\overleftarrow{Y}_t^\theta\}_{t=0}^{T}$ the backward process of DLPM, as introduced in (16). As emphasized in Table 4, the sampling strategies for LIM and DLPM differ fundamentally when $\alpha \neq 2$. This is also the case for the training procedure.

**Stochastic sampling.** The DLPM approach introduces the bounded random variable $0 \leqslant \Gamma_t \leqslant 1$, interacting with the mean and variance of the Gaussian conditional at hand. Three points: when $\alpha = 2$, $\Gamma_t$ becomes deterministic and one recovers DDPM formulas. Second, $\Gamma_t$ brings additional stochasticity in the sampling process. Third, it does so in the interesting manner than it simultaneously scales both (i) the magnitude of the noise added at time $t-1$ and (ii) the output of the noise model.

**Deterministic sampling.** In the case of the scale-preserving schedule, these two equations do not describe the same sampling procedure.

|  | Stochastic | Deterministic |
|---|---|---|
| Continuous (LIM) | $\frac{\overleftarrow{X}_t^\theta}{\gamma_t} - \frac{\alpha(1/\gamma_t - 1)}{\sigma_{1 \to t}^{\alpha-1}} \hat{\epsilon}_t^\theta + (\frac{1}{\gamma_t^\alpha} - 1)^{1/\alpha} \epsilon_t'$ | $\frac{\overleftarrow{X}_t^\theta}{\gamma_t} - \left( \frac{\sigma_{1 \to t}^{1-\alpha}}{\gamma_t} - \sigma_{1 \to t}^{1-\alpha} \right) \hat{\epsilon}_t^\theta$ |
| Denoising (DLPM) | $\frac{\overleftarrow{Y}_t^\theta}{\gamma_t} - \Gamma_t \sigma_{1 \to t} \hat{\epsilon}_t^\theta + \sqrt{\Gamma_t \Sigma_{1 \to t-1}} G_t'$ | $\frac{\overleftarrow{Y}_t^\theta}{\gamma_t} - \left( \frac{\sigma_{1 \to t}}{\gamma_t} - \sigma_{1 \to t-1} \right) \hat{\epsilon}_t^\theta$ |

Table 4: Distribution of $\overleftarrow{X}_{t-1}^\theta, \overleftarrow{Y}_{t-1}^\theta$. $\{G_t'\}_{t=T}^{1}$ are independent random variables distributed as $\mathcal{N}(0, \mathrm{I}_d)$, $\{\epsilon_t'\}_{t=T}^{1}$ are independent random variables distributed as $\mathcal{S}_\alpha^{\mathrm{i}}(0, \mathrm{I}_d)$. $\hat{\epsilon}_t^\theta$ is the model at hand at time $t$, the formula for $\Sigma_{1 \to t}$ is given in (12), and $\Gamma_t = 1 - \gamma_t^2 \Sigma_{1 \to t-1}/\Sigma_{1 \to t}$. Eventhough $\Gamma_t$ involves two heavy-tailed random variables, it is bounded: $0 \leqslant \Gamma_t \leqslant 1$ (see Appendix C.3).

**Training.** Alike the Gaussian case ($\alpha = 2$), the score $S_t^{(\alpha)}(x_t|x_0)$ is a linear expression of the noise term $\epsilon_t(x_t, x_0)$, so the training equations are very similar, and can be reformulated to involve a denoising loss:

$$\mathcal{L}_{t-1} : \theta \mapsto \mathbb{E}\left( \|\hat{\epsilon}_t^\theta(X_t) - \epsilon_t(X_t, X_0)\|_p^\eta \right). \tag{29}$$

- In the case of DLPM, our discussion leads us to the choice $p = 2$ and $\eta = 1$ (see (8)).

- In the case of LIM, the theory must rely on the choice $p = 2$ and $\eta = 2$ in order to obtain the denoising score matching loss equivalence (*i.e.*, $L, L'$ are equivalent in (28)). One must make the assumption that the losses $L, L'$ are not infinite for some $\theta$, which is not necessarily realistic because $S_t(X_t), S_t(X_t|X_0)$ are heavy-tailed random variables involving $\alpha$-stable noise, and as such admit no variance.

- In the case of LIM, in the experiments the parameters $p = 1$ and $\eta = 1$ are chosen, instead of the previous squared loss, in order to obtain more stable training, potentially indicating that indeed $L, L'$ (28) might be infinite.

## F    TECHNICAL RESULTS

In this section, we give the proofs relative to our technique for faster training, as introduced in Appendix C.5.

**Lemma 1.** *Let $\bar{A}_0^t, \bar{A}_1^t$ bet two independent random variables distributed as $\mathcal{S}_{\alpha/2,1}(0, c_A)$. Define $Z_0 = Y_0$, and*

$$Z_t = \gamma_{1 \to t} Z_0 + \sigma_{1 \to t} \left(\bar{A}_1^t\right)^{1/2} G_t .$$

*Moreover, let $Z_{t-1}$ be equal to*

$$Z_{t-1} = \frac{1}{\gamma_t} \left(Z_t - \Gamma'_t(\bar{A}_{0,1}^t)\sigma_{1 \to t}\epsilon_t(Z_t, Z_0)\right) + \Sigma'_t(\bar{A}_{0,1}^t)G_{t-1} ,$$

*where*

$$\Sigma'_t(\bar{A}_{0,1}^t) = \Gamma'_t(\bar{A}_{0,1}^t)\sigma_{1 \to t-1} \left(\bar{A}_0^t\right)^{1/2}$$

$$\Gamma'_t(\bar{A}_{0,1}^t) = \frac{\bar{A}_1^t \sigma_t^2}{\bar{A}_1^t \sigma_t^2 + \gamma_t^2 \sigma_{1 \to t-1}^2 \bar{A}_0^t}$$

$$\epsilon_t(Z_t, Z_0) = \frac{Z_t - \gamma_{1 \to t} Z_0}{\sigma_{1 \to t}} .$$

*Then the joint distribution of $(Z_0, Z_{t-1}, Z_t, \bar{A}_1^t, \bar{A}_0^t)$ matches the joint distribution of*

$$\left(Y_0, Y_{t-1}, Y_t, A_t, \frac{\Sigma'_t(\bar{A}_{0,1}^t)}{\sigma_{1 \to t}^2}\right) .$$

*Proof.* Consider the setting of Proposition 6. The distribution of $Y_{t-1}$ given $Y_t, Y_0, A_{1:T}$ is characterized by the values of $\Sigma_{1 \to t}, \Gamma_t$:

$$\tilde{m}_{t-1} = \frac{1}{\gamma_t} \left(Y_t - \Gamma_t \sigma_{1 \to t}\epsilon_t(Y_t, Y_0)\right), \quad \tilde{\Sigma}_{t-1}(A_{1:t}) = \Gamma_t(A_{1:t})\Sigma_{1 \to t-1}(A_{1:t-1}) ,$$

where

$$\Sigma_{1 \to t}(A_{1:t}) = \sigma_t^2 A_t + \gamma_t^2 \Sigma_{1 \to t-1}(A_{1:t-1})$$

$$\epsilon_t(Y_t, Y_0) = \frac{Y_t - \gamma_{1 \to t} Y_0}{\sigma_{1 \to t}}$$

$$\Gamma_t(A_{1:t}) = 1 - \frac{\gamma_t^2 \Sigma_{1 \to t-1}(A_{1:t-1})}{\Sigma_{1 \to t}(A_{1:t})} .$$

Directly applying the result of Lemma 2, we can affirm that

$$\Sigma_{1 \to t-1}(A_{1:t-1}) \overset{d}{=} \sigma_{1 \to t-1}^2 \bar{A}_0^t ,$$

where $\bar{A}_0^t \sim \mathcal{S}_{\alpha/2,1}(0, c_A)$. In this conditions, the distribution of $\Gamma_t(A_{1:t})$ is equal to that of $\Gamma'_t(\bar{A}_{0,1}^t)$, where

$$\Gamma'_t(\bar{A}_{0,1}^t) = 1 - \frac{\gamma_t^2 \sigma_{1 \to t-1}^2 \bar{A}_0^t}{\sigma_t^2 A_t + \gamma_t^2 \sigma_{1 \to t-1}^2 \bar{A}_0^t}$$

Since the distribution of $Z_t$ does not change if we draw another independent $\bar{A}_1^t$ instead of $A_t$, this ends the proof. $\qquad\square$

**Lemma 2** (Sampling $\Sigma_{1 \to t}$ with a single heavy-tailed r.v). *Consider the setting of the data augmentation approach in Section 3.2.2, where in particular $\{A_t\}_{t=1}^T$ are independent random variables distributed according to $A_t \sim \mathcal{S}_{\alpha/2,1}^1(0, c_A)$, with $c_A = \cos^{2/\alpha}(\pi\alpha/4)$. Consider the random variable $\Sigma_{1 \to t}(A_{1:t})$, as defined in (12):*

$$\Sigma_{1 \to t}(A_{1:t}) = \sum_{k=1}^t \left( \frac{\gamma_{1 \to t}}{\gamma_{1 \to k}} \sqrt{A_k} \sigma_k \right)^2 .$$

*Then*

$$\Sigma_{1 \to t}(A_{1:t}) \overset{d}{=} \sigma_{1 \to t}^2 A,$$

*where $A \sim \mathcal{S}_{\alpha/2,1}^1(0, c_A)$.*

*Proof.* By Proposition 5, $Y_t$ given $Y_0, A_{1:t}$ is a random variable distributed as a Gaussian of variance $\Sigma_{1 \to t}(A_{1:t})$:

$$Y_t \overset{d}{=} \gamma_{1 \to t} Y_0 + \sqrt{\Sigma_{1 \to t}(A_{1:t})} \bar{G}_t ,$$

where $\bar{G}_t$ is distributed as a standard Gaussian. Remember that $Y_t$ and $X_t$ admit the same distribution, with $X_t = \overset{d}{=} \gamma_{1 \to t} X_0 + \bar{\epsilon}_t$ where $\bar{\epsilon}_t$ is distributed as a $\mathcal{S}_\alpha^i(0, \mathrm{I}_d)$.

In the same spirit we can define a third sequence of random variables $\{Z_t\}_{t=0}^T$ with $Z_0 = X_0$, and

$$Z_t = \gamma_{1 \to t} Z_0 + \sigma_{1 \to t} \sqrt{A_t'} \bar{G}_t ,$$

where $\{A_t'\}_{t=0}^T$ are independent random variables distributed according to $A_t' \sim \mathcal{S}_{\alpha/2,1}^1(0, c_A)$. It is then quite clear from Section 2 that $Z_t$ and $Y_t$ admit the same distribution; in particular,

$$\sqrt{\Sigma_{1 \to t}} G_t \overset{d}{=} \sigma_{1 \to t} \sqrt{A_t'} G_t' .$$

From there, we use Lemma 3 to conclude that $\sqrt{\Sigma_{1 \to t}} \overset{d}{=} \sigma_{1 \to t} \sqrt{A_t'}$, which ends the proof. $\square$

**Lemma 3.** *Let $A, A'$ be positive real random variables, let $Z$ be a real continuous random variable with density $p_Z$. Suppose that $AZ$ and $A'Z$ admit the same distribution. Then $A, A'$ admit the same distribution too.*

*Proof.* Let $h$ be a measurable function. Then $\mathbb{E}(h(A)) = \mathbb{E}(h(AZ/Z)) = \mathbb{E}(h(A'Z/Z)) = \mathbb{E}(h(A'))$. This shows that $A, A'$ have the same distribution. $\square$

## G ADDITIONAL EXPERIMENTAL DETAILS

All experiments are conducted using PyTorch. We use linear timesteps during training and sampling, and the scale-preserving process[6], being the only forward process readily provided by LIM. This entails choosing a sequence $\{\beta_t\}_{t=1}^T$ such that

$$\gamma_t = (1 - \beta_t)^{1/\alpha}, \quad \sigma_t = (1 - \gamma_t^\alpha)^{1/\alpha} ,$$

resulting in $\sigma_{1 \to t} = (1 - \gamma_{1 \to t}^\alpha)^{1/\alpha}$ and $\gamma_{1 \to t} = \prod_{i=1}^t \gamma_i$. With this choice, we obtain approximately $X_T \sim \mathcal{S}_\alpha(0, \mathrm{I}_d)$. We choose $\{\beta_i\}_{i=1}^T$ as the cosine schedule, as introduced by Nichol & Dhariwal (2021).

We do not give any of the heavy-tailed random variables $\{A_t\}_{t=1}^T$ as input to the neural network architecture, as we have witnessed worse performance in every scenarios we tried: learned embedding added to each model layer, concatenation to model input, concatenation at each layer, or feeding $\log(A_{1:T})$ instead of $A_{1:T}$ to better manage large jumps. This corresponds to the design choice D2 in Appendix C.4.2.

For image data generation with LIM, we use the same clipping hyper-parameters specified in Yoon et al. (2023).

All the training and experiments are conducted on four NVIDIA RTX8000 GPU and four NVIDIA V100 GPU, where a single training run on MNIST or CIFAR10_LT takes approximately 1 day per GPU, and requires about 4-12GB of VRAM for the batch sizes we use. Generating 5000 images with 1000 backward steps takes approximately 3-4 hours on one RTX8000 GPU.

---

[6] we mention again that it is traditionally called the variance preserving process

### G.1 2D DATA

We give more details about the mixture of Gaussian we consider in our experiment. It is designed in a grid-like pattern in $[-1, 1]^2$, as follows:

$$\sum_{i=1}^{9} w_i \cdot \mathcal{N}(\mu_i, \sigma^2 \mathrm{I}_2) \;,$$

where $(w_i)_{i=1}^{9} = (0.01,\ 0.1,\ 0.3,\ 0.2,\ 0.02,\ 0.15,\ 0.02,\ 0.15,\ 0.05)$, $\mu_i = (\mu_1, \mu_2)$ with $\mu_1 = (i \bmod 3) - 1$, $\mu_2 = \lfloor i/3 \rfloor - 1$, and $\sigma = 0.05$.

For our 2D datasets, we use 32000 datapoints for training, a batch size of 1024, and 25000 points for evaluation. We train each model for 10000 steps. Since we do not focus on the effect of diffusion steps, we set it to 100, where all methods have been observed to perform optimally.

The optimizer is Adam (Kingma & Ba (2017)) with learning rate 5e-3. We use a neural network consisting of four time-conditioned MLP blocks with skip connections, each of which consisting of two fully connected layers of width 64. The time $t$ passes through two fully connected layers of size 32x32, and is fed to each time conditioned block, where it passes through an additional 32x64 fully connected layer before being component-wise added to the middle layer.

We compute a mean squared logarithmic error (MSLE) loss, designed to assess the fit to tails of distributions. Since it depends on the one-dimensional cumulative distribution function, we calculate it after projecting the data onto each dimension. In this simple setting, we keep the score computed on the first dimension.

We also compute the precision/recall metrics, as presented in Appendix G.3.

### G.2 IMAGE DATA

We work on the MNIST and the CIFAR10_LT dataset. CIFAR10_LT consists of the CIFAR10 images were artificial class unbalance has been introduced. The specific class counts we use are $[5000, 2997, 1796, 1077, 645, 387, 232, 139, 83, 50]$.

The optimizer is Adam (Kingma & Ba (2017)) with learning rate 1e-3 for MNIST and 2e-4 for CIFAR10_LT. We use the StepLR scheduler which scales the learning rate by $\gamma = .99$ every $N = 1000$ steps for CIFAR10_LT and $N = 400$ for MNIST.

To establish a fair comparison, LIM and DLPM use the same network model. We use a U-Net following the implementation of Nichol & Dhariwal (2021) available in `https://github.com/openai/improved-diffusion`. We dimension the network as follows: we set the hidden layers to $[128, 256, 256, 256]$, fix the number of residual blocks to 2 at each level, and add self-attention block at resolution 16x16, using 4 heads. We use an exponential moving average with a rate of 0.99 for MNIST and 0.9999 for CIFAR10_LT. We use the silu activation function at every layer. Diffusion time $t$ is rescaled to $(0, 1)$ and fed to the model through the Transformer sinusoidal position embedding (Vaswani et al. (2023)). We train MNIST for 120000 steps with batch size 256 with a time horizon $T = 1000$, and CIFAR_LT for 400000 steps with batch size 100 with a time horizon $T = 4000$.

We use the FID metric for assessing the quality of our generative models, computing this metric between 5000 using images and 5000 generated images.

### G.3 METRICS FOR GENERATIVE MODELS

**MSLE** we use a mean squared logarithmic error (MSLE) metric tailored to measure the fit on the tails of the distribution at hand. Drawing inspiration from Allouche et al. (2022), we define the MSLE as the squared distance between the logarithm of the inverse cumulative distributions of the original and generated data. If $F, \hat{F}$ denote respectively the cumulative distribution function of the original data and the empirical cumulative distribution function of the generated data, then

$$\mathrm{MSLE}(\xi) = \int_{\xi}^{1} \left( \log F^{-1}(p) - \log \hat{F}^{-1}(p) \right)^2 dp \;,$$

where $\xi$ is chosen the be 0.95.

**Precision/recall**  These metrics are introduced in the setting of generative models by Sajjadi et al. (2018), and assess the overlap of sample distributions using local geometric structures. Precision measures how much the generated distribution is contained in the original data distribution (measuring quality), and recall measured how much of the original data distribution is covered by the generated distribution (diversity). We also consider the $F_1^{\text{pr}}$ score which we define as the harmonic mean of these two values:

$$F_1^{\text{pr}} = 2 \cdot \frac{\text{precision} \cdot \text{recall}}{\text{precision} + \text{recall}} \ .$$

## G.4  ADDITIONAL RESULTS

We provide some more results on MNIST and CIFAR10_LT, with FID for non-isotropic noise, and with the $F_1^{\text{pr}}$ metric for other methods (with clipping enabled in LIM and LIM-ODE). We also provide grid images in order to visually check the performance of DLPM.

| MNIST | $\alpha = 1.5$ | $\alpha = 1.7$ | $\alpha = 1.8$ | $\alpha = 1.9$ | $\alpha = 2.0$ |
|---|---|---|---|---|---|
| DLPM$^{\text{ni}}$ | 44.17 | 14.06 | 5.74 | 3.62 | - |
| DLIM$^{\text{ni}}$ | 14.96 | 51.58 | 59.84 | 76.03 | - |

Table 5: FID↓, 1000 sampling steps for DLPM$^{\text{ni}}$, 25 sampling steps for DLIM$^{\text{ni}}$.

| | DLIM | DLPM | LIM | LIM-ODE | DDPM |
|---|---|---|---|---|---|
| $\alpha$ | | | | | |
| 1.7 | 0.884 | **0.887** | 0.857 | 0.869 | - |
| 1.8 | 0.874 | **0.881** | 0.821 | 0.875 | - |
| 1.9 | 0.877 | **0.878** | 0.700 | 0.808 | - |
| 2.0 | 0.820 | 0.871 | 0.694 | 0.772 | **0.881** |

Table 6: MNIST, $F_1^{\text{pr}} \uparrow$

| | DLIM | DLPM | LIM | LIM-ODE | DDPM |
|---|---|---|---|---|---|
| $\alpha$ | | | | | |
| 1.7 | 0.676 | 0.675 | **0.679** | 0.677 | - |
| 1.8 | 0.669 | **0.680** | 0.677 | 0.673 | - |
| 1.9 | 0.667 | **0.669** | 0.661 | 0.669 | - |
| 2.0 | 0.664 | **0.667** | 0.660 | 0.665 | 0.666 |

Table 7: CIFAR10_LT, $F_1^{\text{pr}} \uparrow$

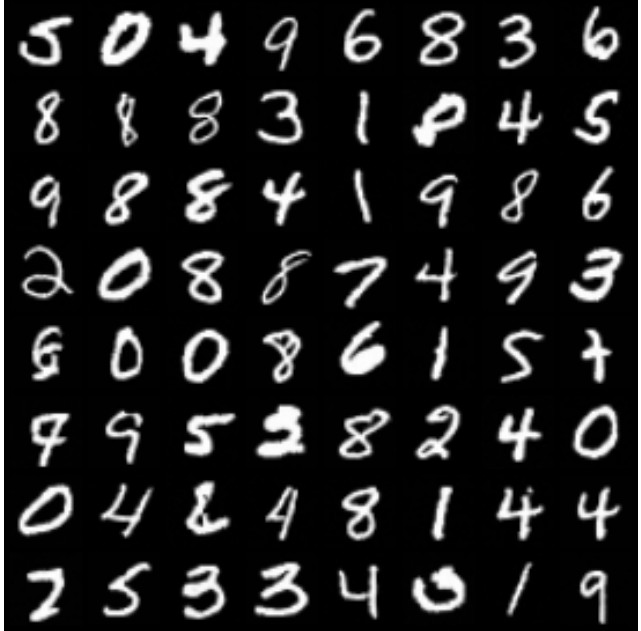

Figure 6: MNIST, DLPM ($\alpha = 1.7$)

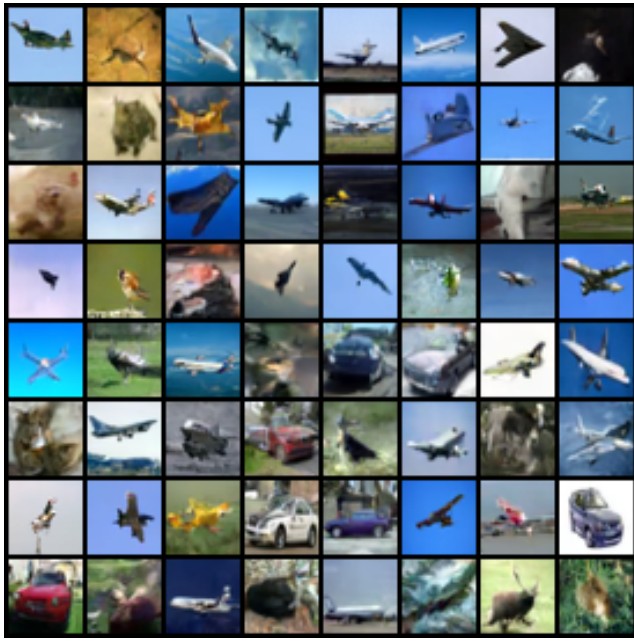

Figure 7: CIFAR10_LT, DLPM ($\alpha = 1.7$)

