# OpenReview forum: "Heavy-Tailed Diffusion with Denoising Levy Probabilistic Models"
_ICLR.cc/2025/Conference — ICLR 2025 Poster_

### Official Review · Reviewer_zxJx · 2024-10-21

**Soundness:** 3
**Presentation:** 3
**Contribution:** 3
**Rating:** 6
**Confidence:** 3

**Summary:**

This paper leverage a intrigue design, decomposing a heavy-tailed r.v. into a Gaussian and a standard alpha-stable r.v. , to propose a new heavy-tail diffusion model with minor modifications to the DDPM. Experiments showed its superiority.

**Strengths:**

1. The math derivation of the alpha-stable dm loss is rigorous.
2. This way of building heavy-tail DM is easier to understand and implement, because DLPM only need sampling to the heavy-tail distribution, there is no need to combine the heavy-tail distribution, into the network.
3. It is novel to make alpha-stable DM solvable via a decomposition.

**Weaknesses:**

1. I have a doubt on the convergence property, for t>0, the marginal distribution of this new dynamic should have infinite variance. Then really it would converge to the data distribution (with finite distribution) at t=0? Can you show me more theoretical convergence proof?
2. The term 'Data Augmentation' typically means increase the size of the dataset by creating new data through modifications or transformations of the original data. I am quite confused why the technique in section 3.2.2 is called 'Data Augmentation'.

**Questions:**

see Weaknesses

---

> ### Author Response · Authors · 2024-11-22
> **Rebuttal**
>
> We thank the reviewer for their thoughtful feedback and for recognizing the strengths of our work. Below, we address the reviewer's concerns in detail.
>
> * *I have a doubt on the convergence property, for t>0, the marginal distribution of this new dynamic should have infinite variance. Then really it would converge to the data distribution (with finite distribution) at t=0? Can you show me more theoretical convergence proof?*
>
> This is a legitimate question raised by the reviewer, to which we can provide a positive answer. To be more precise, based on the work in [1, 2, 3] mentioned in the introduction of our paper (line 81), we can consider the following Lévy-driven SDE:
> $$
>     d X_t = b(X_t) dt + \sigma dL_t^{\alpha}, \quad t\geq 0.
> $$
> For an appropriate choice of $b$, the limiting distribution of $X_t$ when $t \to \infty$ can have finite variance. Similarly a Brownian SDE defined by
> $$
>     d X_t = b(X_t) dt + \sigma(X_t) dB_t, \quad t \geq 0,
> $$
> can have a limiting distribution with infinite variance with appropriate choice of $b$ and $\sigma$ [4].
>
> Therefore, the tails of the noise do not always determine the tails of the limiting distribution, and it is indeed possible to converge to the true data distribution.
>
> For our specific work, which relies on a discrete time structure, we can draw inspiration from existing works that establish theoretical guarantees for DDPM, particularly [5, 6].
>
> We are confident that by adapting the approaches presented in these studies, we will be able to derive convergence bounds for DLPM. However, we would like to emphasize that these works on DDPM are relatively recent and involve complex, sophisticated techniques. Therefore, we believe that providing a complete answer to the issue you raised is beyond the scope of this paper and is best reserved for future work.
>
>
> [1] Simsekli et al., 2017, Exploring Levy driven stochastic differential equations for Markov chain Monte Carlo
>
> [2] Simsekli et al., 2020, Underdamped langevin dynamics: Retargeting sgd with momentum under heavy-tailed gradient noise
>
> [3] Huang et al., 2020, Approximation of heavy-tailed distributions via stable-driven sdes
>
> [4] Bresar et al., 2024, Subexponential lower bounds for f-ergodic Markov processes
>
> [5] Li, G. and Yan, Y, 2024, O(T/d) convergence theory for diffusion probabilistic models under minimal
> assumptions.
>
> [6] Z Huang, et al., 2024, Denoising diffusion probabilistic models are optimally adaptive to unknown low dimensionality
>
> * *The term 'Data Augmentation' typically means increase the size of the dataset by creating new data [...]. I am quite confused [...]*
>
> The term "Data Augmentation" originates from the Markov Chain Monte Carlo (MCMC) literature, as referenced in Chapter 10 of [1]. This legacy has influenced other popular machine learning methods, such as Augmented Neural ODEs [2].
>
> We acknowledge the potential for confusion due to the differing interpretation of the term in the deep learning context. We will clarify this distinction in the revised paper to avoid ambiguity.
>
> [1] Chapman et al., 2011, Handbook of Markov Chain Monte Carlo
>
> [2] Dupont et al., 2019, Augmented Neural ODEs

---

> ### Author Response · Authors · 2024-11-25
> **Reminder: Rebuttal Discussion Period Ending Soon**
>
> Dear Reviewer,
>
> We hope this message finds you well. We are writing to kindly remind you that the discussion period for the review process will soon come to an end. As the authors, we wanted to ensure that our rebuttal has been received and addressed.
>
> In particular, we would greatly appreciate your feedback or acknowledgment on the following key points raised in our rebuttal:
>
> * **Convergence property**: We provided clarifications on the marginal distribution of stochastic processes driven by light-tailed or heavy-tailed dynamics, and how these tails do not always determine the tails of the limiting distribution. This makes it possible to converge to the true data distribution.
>
> * **Terminology (Data augmentation)**: We clarified our choice for the term 'Data augmentation', which originates from the MCMC literature, and which has been used in other popular machine learning methods.
>
> If there are any remaining questions or points requiring clarification, we would be happy to provide additional responses or discuss further. Your feedback and engagement are invaluable to improving the quality of our work. Thank you very much for your time and contributions to the review process.
>
> Best regards,
>
> The authors

---

> > ### Comment · Reviewer_zxJx · 2024-11-27
> >
> > Thanks for your reply, my concerns are addressed.
> >
> > I think the methodology of this work is novel.
> > I will keep my score as 6.
> >
> > However, it seems other reviewers have concerns about experiments, I am not very familiar with these. You can concentrate on discussion with them. If they are convinced that the experiments are solid, I would like to back up your acceptence.

---

> > > ### Author Response · Authors · 2024-11-27
> > >
> > > Dear reviewer,
> > >
> > > Thank you for your thoughtful reply and for considering the novelty of our methodology. We greatly appreciate your willingness to support our work and your constructive feedback throughout this process.
> > >
> > > Best regards,
> > >
> > > The authors

---

### Official Review · Reviewer_NXD4 · 2024-11-02

**Soundness:** 3
**Presentation:** 3
**Contribution:** 3
**Rating:** 6
**Confidence:** 3

**Summary:**

This work studies the diffusion-based models with a heavy-tailed noise and achieves great performance under the imbalanced, heavy-tailed data distribution. Different from the previous LIM method, which starts from a continuous SDE perspective, this work follows the DDPM framework and proposes DLPM and DLIM models. For the proposed methods, this work also shows a suitable training objective to achieve stable training for the $\alpha$-stable distribution. The synthetic and real-world experiments also support their results.

**Strengths:**

1. The link between the $\alpha$-stable distribution and Gaussian noise is helpful since we can adapt the DDPM framework by a simple transformation.
2. The suitable training objective of DLPM bridges the gap between theory and application. On the contrary, the LIM needs to choose a $L_1$ loss to guarantee stable training.

**Weaknesses:**

It would be better to discuss the experiment results in detail. Please see the Question part to find the details.

**Questions:**

Q1: When considering CIFAT10_LT, LIM performs better than the DLPM method, which can not support the discussion of this work. Could the author discuss it in detail?

Q2: As shown in the theoretical part, if $\alpha=2$, the DLPM algorithm is equivalent to DDPM. However, the experiment results of DLPM ($\alpha=2$) are worse than DDPM. Could the author discuss this experiment phenomenon in detail?

Q3: Could the author discuss the reason for the slightly worse performance of $DLPM^{\text{in}}$?

---

> ### Author Response · Authors · 2024-11-22
> **Rebuttal**
>
> We greatly thank the reviewer for their comments, and we address their concerns below.
>
> * *Q1: When considering CIFAT10\_LT, LIM performs better than the DLPM method, which can not support the discussion of this work. Could the author discuss it in detail?*
>
> While our results demonstrate the strengths of DLPM, we acknowledge that  in some cases  LIM shows better performance. This is due to its use of clipping techniques, which we intentionally did not include in our method. Without these clipping adjustments, LIM’s performance significantly degrades, as we see in the updated Table 3 given below. This highlights that its framework is inherently less suited for heavy-tailed noise. Our approach emphasises simplicity and theoretical grounding, providing an "out-of-the-box" method without requiring extensive hyperparameter tuning.
>
> **Table 3: FID$\downarrow$, 1000 sampling steps for LIM, 25 sampling steps for LIM-ODE.**
>
> | MNIST   | α=1.5 | α=1.7 | α=1.8 | α=1.9 | α=2.0 |
> |--|--|--|--|-|--|
> | LIM      | 14.37 | 11.54 | 11.18 | 13.75 | -     |
> | LIM-ODE  | 49.63 | 78.59 | 92.93 | 109.48| -     |
>
> | CIFAR10_LT   | α=1.5 | α=1.7 | α=1.8 | α=1.9 | α=2.0 |
> |--|-|--|--|---|-|
> | LIM      | 75.38 | 35.15 | 31.14 | 21.68 | -     |
> | LIM-ODE  | 42.07 | 91.64 | 105.95| 407.79| -     |
>
> We anticipate that incorporating additional techniques like clipping into DLPM would yield further improvements, but we leave these further methodological refinements for future work.
>
> * *Q2: As shown in the theoretical part, if $\alpha = 2$, the DLPM algorithm is equivalent to DDPM. However, the experiment results of DLPM ($\alpha=2$) are worse than DDPM. Could the author discuss this experiment phenomenon in detail?*
>
> As we mention in footnote $4$ of page 6, the DLPM algorithm with $\alpha = 2$ exactly recovers  DDPM when we use $r=1$ in our loss (7).
>
> However, in our experiments, we used $r=1/2$ in our loss, as a default setting. This choice was made for simplicity, as it provides a loss formulation that works across all $\alpha$ values.
> This default choice is of course less suitable for $\alpha = 2$, when all considered distributions have finite variance, leading to the observed discrepancy in performance when compared to DDPM.
>
> An alternative approach could involve choosing $r$ dynamically, such as $r = \frac{\alpha - (2 - \alpha)c}{2}$ (with $c < 1$), which interpolates between our loss and the standard Gaussian loss, or simply setting $r = 1/2$ if $\alpha < 2$, and $r=1$ if $\alpha=2$.
>
> If the reviewer feels this is an important point, we will include this discussion in the revised paper.
>
> * *Q3: Could the author discuss the reason for the slightly worse performance of DLPM$^{\text{ni}}$?*
>
> Several factors may contribute to this phenomenon:
> * The loss function is better suited for isotropic noise distributions
> * Neural network architectures may inherently favour isotropic noise (e.g., due to rotational invariance in $\ell_2$ space)
> * The datasets considered in our experiments might align better with isotropic noise assumptions
>
> While non-isotropic noise does not outperform isotropic noise in our experiments, it could be a promising approach in other datasets or contexts. We consider this a methodological question that warrants further investigation.

---

> ### Author Response · Authors · 2024-11-25
> **Reminder: Rebuttal Discussion Period Ending Soon**
>
> Dear Reviewer,
>
> We hope this message finds you well. We are writing to kindly remind you that the discussion period for the review process will soon come to an end. As the authors, we wanted to ensure that our rebuttal has been received and addressed.
>
> In particular, we would greatly appreciate your feedback or acknowledgment on the following key points raised in our rebuttal:
>
> * **Generative Performance of DLPM vs. LIM**: We provided additional clarification on LIM’s reliance on clipping and shared results from new experiments without clipping, which highlight the robustness and suitability of DLPM for heavy-tailed settings.
>
> * **Clarification on DDPM vs DLPM**: We clarified how our loss is constructed and thus how DLPM compares to DDPM when $\alpha < 2$ and when $\alpha=2$.
>
> * **Clarification on DLPM$^{\text{ni}}$**: We further discussed the performance of our algorithm with non-isotropic noise, which could be a promising approach in other datasets or contexts.
>
> If there are any remaining questions or points requiring clarification, we would be happy to provide additional responses or discuss further. Your feedback and engagement are invaluable to improving the quality of our work. Thank you very much for your time and contributions to the review process.
>
> Best regards,
>
> The authors

---

> > ### Comment · Reviewer_NXD4 · 2024-11-26
> >
> > Thanks for the detailed reply. Most of my concerns are addressed. Can you provide the clipping version of your algorithm? I think this will make the experiment more complete.

---

> > > ### Author Response · Authors · 2024-11-27
> > >
> > > Dear reviewer,
> > >
> > > Thank you for your detailed feedback and for acknowledging that most of your concerns have been addressed.
> > >
> > > Regarding your suggestion to provide results for the clipping version of our algorithm, we agree that such results would enhance the completeness of the experimental section. Even though we are limited by our computational budget, we will do our best to provide these results as soon as possible, if feasible within the timeline of the review process.
> > >
> > > We appreciate your understanding and your constructive input, which has helped us improve our work.
> > >
> > > Best regards,
> > >
> > > The authors

---

### Official Review · Reviewer_o74J · 2024-11-02

**Soundness:** 2
**Presentation:** 3
**Contribution:** 3
**Rating:** 6
**Confidence:** 3

**Summary:**

In the context of generating data from a heavy-tailed distributions, the authors introduce a new generative model termed the Denoising Levy Probabilistic Model (DLPM). Previous work by Yoon et al (2023) had introduced the Levy-Ito Model (LIM) which is an extension of the SDE from Score-Based Diffusion Models (SBDM) by Song et al (2021) to the case of heavy-tailed distributions, where the noise added at each step follows an $\alpha$-stable distribution. On the other hand, the authors extend the Denoising Diffusion Probabilistic Model (DDPM) from Ho et al (2020) to the case of heavy-tailed distributions. It is well known that DDPM is equivalent to a particular discretization of the Variance Preserving SDE from Song et al (2021). However, when extended to heavy-tailed distributions, LIM and DLPM are not equivalent. In fact, the framework of DLPM is simpler and more principled than the LIM and allows for reusing previous code implementations of DDPM. A deterministic sampler counterpart to DLPM is introduced and showed to outperform the deterministic counterpart of LIM when few sampling steps are considered in experiments. More experiments are provided comparing the different models with mixed success.

**Strengths:**

* The presentation is clear and the necessary background is succesfully introduced.  There is a good flow on building the DLPM as a natural extension of the DDPM.
* The provided framework for DLPM has the nice property that it has DDPM as special case. More precisely, DLPM at each noising step adds a sample from a $\alpha$-stable distribution. In the case of $\alpha=2$, DLPM reduces to DDPM (except for a square root in the loss.) This is mathematically interesting and puts DDPM into a more general and flexible framework.
* The authors do a good job comparing DLPM with LIM. In particular, DLPM uses more elementary tools and a more clear presentation than LIM. Yoon et al (2023) develop the theory for the LIM framework using an $\ell_2$ loss while the experiments are done using an $\ell_1$ loss. In contrast, the model for DLPM that is theoreticallly studied is the same as the one used for the experiments.
* The reparameterization provided in assumption D2 ensures that the function that the neural network estimates does not depend explicitly on the $\alpha-$stable distribution which means that previous implementations for DDPM can be reused.

I find that their main contribution is to provide a better and simplified mathematical framework for diffusion when the noise added can have heavy-tails.

**Weaknesses:**

I do not find the claim that the authors make on the experiments as satisfactory as the theoretical counterpart. Especially in proposing a new framework (for a task that had an already existing framework, i.e., LIM) it is important to ask whether the new framework leads to better performance in a class of tasks.

1) The experiment for CIFAR10_LT is not conclusive: we see in Table 3 that in terms of FID, LIM is no worse than DLPM and for most $\alpha$ LIM actually outperforms DLPM. On the other hand, DLIM does outperform LIM-ODE measured by FID, but only when taking 25 sampling steps. The plot in Figure 4 shows that taking 100 steps or more LIM-ODE is comparable or outperforms DLIM. The only claim that could be drawn from here is that DLIM (the deterministic sampler for DLPM) is better when constrained to a few number of steps than LIM-ODE (the deterministic sampler for LIM.) Even this is not supported by further experimental data provided in the appendix, Table 6, where it is shown that LIM-ODE has better $F_1^\text{pr}$ scores than DLIM for all $\alpha.$ A more fair claim is that the performance of DLPM/DLIM is comparable to that of LIM/LIM-ODE.

2) Also, in Table 4 we see that the FID for DLPM at $\alpha=2$ is 21.07 while the FID for DDPM is 19.05. This is a big difference and seems to weaken the point that DLPM at $\alpha=2$ is similar to DDPM. We know DDPM and DLPM at $\alpha=2$ are different since one has a square root in the definition of the loss.

3) It seems to me important to consider $\alpha=1.5$ for the experiments whose results are shown in Table 1 and Table 2. Indeed, Yoon et al (2023) work in the interval $\alpha \in [1.5, 2]$ and we further see that performance for LIM is clearly increasing as $\alpha$ decreases both for the results in Table 1 and Table 2.

4) It is claimed in the paragraph above Table 2 that DLPM5 has a better performance than DLPM over all the range of $\alpha$ for data coming from a Gaussian grid but a t-test shows that this difference is not statistically significant for $\alpha=1.7$ and $\alpha=1.8.$ Also, the results reported in Table 6 for $\alpha=\{1.7,1.8,1.9\}$ do not seem to be statistically significant for supporting the claim that DLPM outperforms LIM, however, standard deviations are not provided.

5) It also seems the experiments missed an important comparison with the ODE counterpart of DDPM, the DDIM from Song et al (2020). That is, even if DLIM outperforms LIM-ODE, it is still relevant to compare it with the non-heavy-tailed version of the ODE framework, which is DDIM.

If this concerns about the experiments are addressed, it is likely I will increase the score.

Smaller comments
* It is not clear why the results for the non-isotropic noise case are mentioned since they give consistently worse results. It may be better to just mention that non-isotropic noise does not provide better performance and push the results about it to the appendix. One advantage of DLPM mentioned is that it can handle the non-isotropic noise, but as said before, it does not seem to give better results.
* line 383 it says 'appart' it should say 'apart.'

**Questions:**

The fact that the neural network does not have the $\alpha-$stable variable as input seems a bit magical. I may be missing something, but it seems that we build all this framework considering the $\alpha-$stable noise but then it only appears implicitly through $Y_t$ and in the conditioning in the loss. I think the reader would benefit from a bit more explanation regarding where is the $\alpha-$stable noise appearing even though the neural network does not model it.

What is behind the empirically seemingly big difference between DDIM at $\alpha=2$ and DDPM mentioned in Weaknesses item (2)?

Could you provide the results for results for $\alpha=1.5$ and $\alpha=1.6$ available for the experiments with results depicted in Table 1 and Table 2? (c.f. the comment in Weaknesses item (3).)

---

> ### Author Response · Authors · 2024-11-22
> **Rebuttal**
>
> We would like to warmly thank the reviewer, who has taken great time and effort to provide in-depth comments. We address their concerns below.
>
> * *The experiment for CIFAR10_LT is not conclusive: [...] A more fair claim is that the performance of DLPM/DLIM is comparable to that of LIM/LIM-ODE. the experiments missed an important comparison [to] DDIM*
>
> As shown in Figure 4, DLPM demonstrates superior performance compared to LIM at smaller iteration counts, which aligns with one of the key advantages of heavy-tailed diffusions: improved efficiency in low-step sampling scenarios.
>
> Regarding the generative performance of DLPM vs LIM with 1000 sampling steps, we appreciate the reviewer’s suggestion and acknowledge the need for additional clarification. In our original submission, our intent was not to present LIM in an overly negative light, so we did not mention the following point.
>
> LIM relies on gradient and noise clipping, which introduces extra hyperparameters that must be fine-tuned for each dataset. To provide a more equitable comparison, we conducted a new set of experiments where LIM was tested without clipping. The results in table below show that LIM’s performance deteriorates significantly, raising questions about whether the framework of LIM is inherently well-suited for heavy-tailed distributions. This highlights a limitation in its robustness relative to DLPM.
>
> **Table 3: FID$\downarrow$, 1000 sampling steps for LIM, 25 sampling steps for LIM-ODE, DDIM.**
>
> | MNIST   | α=1.5 | α=1.7 | α=1.8 | α=1.9 | α=2.0 |
> |--|--|--|--|-|--|
> | DDIM     |  |   |    |   | 5.16  |
> | LIM      | 14.37 | 11.54 | 11.18 | 13.75 | -     |
> | LIM-ODE  | 49.63 | 78.59 | 92.93 | 109.48| -     |
>
> | CIFAR10_LT   | α=1.5 | α=1.7 | α=1.8 | α=1.9 | α=2.0 |
> |--|-|--|--|---|-|
> | DDIM     |   |   |   |   | 23.44 |
> | LIM      | 75.38 | 35.15 | 31.14 | 21.68 | -     |
> | LIM-ODE  | 42.07 | 91.64 | 105.95| 407.79| -     |
>
> We would like to add that if we applied such clipping techniques, we would certainly achieve improved results. However, our paper's philosophy prioritizes simplicity and theoretical rigor over optimization-focused interventions.
>
> Regarding DDIM, we agree that this is an important comparison. We have reported its performance in the same table below. We will include these results and remarks in the paper revision.
>
> * *We see that the FID for DLPM at $\alpha=2$ is 21.07 while the FID for DDPM is 19.05. This is a big difference and seems to weaken the point that DLPM at $\alpha=2$ is similar to DDPM.*
>
> As we mention in footnote $4$ of page 6, the DLPM algorithm with $\alpha = 2$ exactly recovers  DDPM when we use $r=1$ in our loss (7).
>
> However, in our experiments, we used $r=1/2$ in our loss, as a default setting. This choice was made for simplicity, as it provides a loss formulation that works across all $\alpha$ values. This default choice is of course less suitable for $\alpha = 2$, when all considered distributions have finite variance, leading to the observed discrepancy in performance when compared to DDPM.
>
> * *Add experiments with $\alpha=1.5, 1.6$ in Table 1 and 2, statistical significance in Table 2*
>
> Again, we would like to thank the reviewer and his attention to details. We have updated Tables 1 and 2 with results from new experiments and computed $p$-values using Welch’s $t$-test to compare the means of DLPM with those of other methods. With the usual threshold of $p < 0.05$, the results are now statistically significant across all settings, except for two cases: when $\alpha = 2$, which is expected since all methods are comparable, and for DDPM vs DLPM$_5$ at $\alpha = 1.7$ ($p = 0.074$). These updates strengthen the evidence supporting our claims.
>
> Upon review, we realised that, for Table 2, we only used 5,000 samples for testing. This limited sample size introduced high variance in precision and recall metrics, especially for the least-represented class, which accounts for only 1% of the samples. Since precision and recall metrics in low dimensions are sensitive to rare class occurrences, we increased the number of test samples to 25,000.
>
> For image data, we currently lack the computational budget required to compute rigorous error bounds. However, we believe that considering the performance of unclipped LIM provides an unambiguous baseline for comparison.
>
> **Table 1: $\text{MSLE}_{\xi = 0.95} \downarrow$, averaged over 20 runs. $p$-values computed using Welch's $t$-test (unequal variance) between means of DLPM and each other method**
>
> | Method | α=1.5       | α=1.6       | α=1.7       | α=1.8 | α=1.9| α=2.0 |
> |--|-|-|--|--|-|--|
> | DLPM   | 0.160 ± 0.128 | 0.081 ± 0.078 | 0.071 ± 0.028 | 0.099 ± 0.044 | 0.132 ± 0.101 | 0.798 ± 0.601 |
> | DDPM   | - | - | - | - | - | 0.528 ± 0.400 |
> |  |    |  | |  |  | *1.0e-1*  |
> | LIM    | 0.743 ± 0.290 | 0.497 ± 0.311 | 0.267 ± 0.077 | 0.653 ± 0.413 | 2.444 ± 1.067 | 1.239 ± 0.240 |
> | | *1.0e-08*   | *8.6e-06*   | *1.3e-10*   | *8.8e-06*   | *7.9e-09*   | *5.0e-3*    |

---

> ### Author Response · Authors · 2024-11-22
> **.**
>
> **Table 2: $F_1 \uparrow$ score, averaged over 30 runs. $p$-values computed using Welch's $t$-test (unequal variance) between means of DLPM and each other method**
>
> | Method   | α=1.5      | α=1.6      | α=1.7      | α=1.8      | α=1.9      | α=2.0      |
> |--|---|--|---|--|---|--|
> | DLPM     | 0.933±0.018 | 0.923±0.005 | 0.933±0.028 | 0.923±0.024 | 0.907±0.034 | 0.862±0.028 |
> | DDPM     | -      | -     | -   | -    | -   | 0.867±0.029 |
> |     |    |    |   |    |    | *5.0e-1*   |
> | DLPM$_5$ | 0.944±0.013 | 0.943±0.021 | 0.943±0.010 | 0.941±0.014 | 0.928±0.016 | -     |
> |          | *9.0e-3*   | *1.6e-05*  | *7.4e-2*   | *9.0e-4*   | *3.9e-3*   |      |
> | LIM      | 0.842±0.039 | 0.850±0.046 | 0.868±0.034 | 0.874±0.030 | 0.884±0.017 | 0.874±0.027 |
> |          | *1.7e-14*  | *1.3e-09*  | *5.7e-11*  | *3.9e-09*  | *1.9e-3*   | *9.6e-2*   |
>
> * *It is not clear why the results for the non-isotropic noise case are mentioned*
>
> Since one of the paper’s main goal is exploration, we felt it was important to assess the performance of non-isotropic noise, which is a natural extension of our framework. Although this variant performs worse for our datasets, it may prove useful for other datasets or tasks.
>
> We will follow the reviewer’s suggestion, and move these results to the appendix in the revised paper.
>
> * *The fact that the neural network does not have the stable variable as input seems a bit magical.*
>
> This is a great observation. In some sense this is consistent with Gaussian diffusion models, where the network takes only the noised data $X_t$ as input at each time $t$; in the classical denoising diffusion interpretation, the role of the network is then to infer the noise $G_t$ implicitly present in $X_t$.
>
> As we mention in Appendix G (line 1556), we have actually run a lot of experiments with the network taking $A_t$ as input in a few different ways, but the performance did not improve.
>
> If the reviewer believes it would benefit the reader, we will expand on it in the main text.

---

> ### Author Response · Authors · 2024-11-25
> **Reminder: Rebuttal Discussion Period Ending Soon**
>
> Dear Reviewer,
>
> We hope this message finds you well. We are writing to kindly remind you that the discussion period for the review process will soon come to an end. As the authors, we wanted to ensure that our rebuttal has been received and addressed.
>
> In particular, we would greatly appreciate your feedback or acknowledgment on the following key points raised in our rebuttal:
>
> * **Generative Performance of DLPM vs. LIM**: We provided additional clarification on LIM’s reliance on clipping and shared results from new experiments without clipping, which highlight the robustness and suitability of DLPM for heavy-tailed settings.
>
> * **Statistical Significance**: We included updated $p$-values using Welch’s $t$-test for key results, addressing concerns about significance for some $\alpha$ values like 1.7 and 1.8.
>
> * **Clarification on DDPM vs DLPM**: We clarified how our loss is constructed and compares to DDPM when $\alpha < 2$ and when $\alpha=2$.
>
> * **Additional results (DDIM, $\alpha=1.5, 1.6$)**: We included DDIM in our updated experiments, and additional results on the 2d datasets for $\alpha=1.5, 1.6$. We welcome any further comments on these additional comparisons.
>
> If there are any remaining questions or points requiring clarification, we would be happy to provide additional responses or discuss further. Your feedback and engagement are invaluable to improving the quality of our work. Thank you very much for your time and contributions to the review process.
>
> Best regards,
>
> The authors

---

> > ### Comment · Reviewer_o74J · 2024-11-27
> >
> > Dear authors,
> >
> > I am sorry but I cannot find the updated version of the paper. You said that you updated Tables 1 and 2, but I just download again the paper and it seems to have the same data as the original paper. Am I missing something?

---

> > > ### Author Response · Authors · 2024-11-27
> > >
> > > Dear reviewer,
> > >
> > > We have now updated the tables in the uploaded pdf (updated Table 1,2 with DDPM and $\alpha=1.5,1.6$ - updated Table 2 with LIM no-clip, using more samples to evaluate the metric, and added p-values - updated Table 3 with LIM no clip and DDIM.)
> > >
> > > Due to the significant effort required to run new experiments and draft the reviews, and since the original deadline for paper changes was yesterday (changed to today), we only provided updates in the rebuttal comments.
> > >
> > > We want to reassure you that potential further changes, if requested by reviewers, will be fully incorporated into the camera-ready version.
> > >
> > > Thank you immensely for your time and feedback.
> > >
> > > Best regards,
> > >
> > > The authors

---

> > > > ### Author Response · Authors · 2024-12-01
> > > >
> > > > Dear Reviewer o74J,
> > > >
> > > > We thank you again for your time and effort in reviewing our paper.
> > > >
> > > > We have provided a detailed rebuttal addressing your concerns. While the discussion stage is approaching its extended deadline, we have not heard back from you. We are sincerely looking forward to your feedback for our rebuttal, particularly as you mentioned that you would reconsider your rating if concerns about experiments were addressed.
> > > >
> > > > Thanks for your continued engagement,
> > > >
> > > > Best Regards,
> > > >
> > > > The Authors

---

> > > > > ### Comment · Reviewer_o74J · 2024-12-02
> > > > >
> > > > > W1. I am still not convinced by the claim that DLIM outperforms LIM-ODE experimentally, especially because of the experiments in Table 6. In any case, DLIM performs at least comparatively to LIM-ODE, and I agree with the authors that DLIM has a simplified framework and does not require of tuning techniques like gradient clipping.
> > > > > W3. Addressed
> > > > > W4. I still believe the results of DLPM5 performing better than DLPM in table 2 are quite weak.
> > > > > W5. Addressed
> > > > >
> > > > > Q1. I missed the paragraph in Appendix G regarding the experiments adding the noise as input to the network. This is interesting, and it addresses my concern.
> > > > > Q2/W2: Addressed
> > > > >
> > > > > Overall, I believe this paper provides a solid framework, and I raise my score to 6. It would be better if the empirical claims in the paper are not amplified and stick with what the data says.

---

### Official Review · Reviewer_Eq2q · 2024-11-09

**Soundness:** 3
**Presentation:** 3
**Contribution:** 2
**Rating:** 6
**Confidence:** 3

**Summary:**

DDPMs degrade images by adding Gaussian noise. This paper proposes substituting Gaussian noise with alpha-stable random variables and defines DLPMs, which are diffusion models with heavy-tailed noise. As the authors solely focus on the discrete-time setting, their exposition is simpler and easier to understand than a previous already existing extension of diffusion models to heavy-tailed noise.

**Strengths:**

The paper is clear in its exposition and makes a good effort to clarify its difference with closely related prior work (Yoon et al). The derivations to make the training loss implementable are interesting.

**Weaknesses:**

1. The training loss is unstable, poorer approximations perform better: The authors overcome the bottleneck of not having analytical KL divergences available through conditioning. Under this conditioning, the denoising network still has to learn the conditional expectation of a heavy-tailed distribution contrary to what is said in the paper. In practice, in order to be implementable, the conditioning requires a Monte Carlo approximation of an expectation. In the experiments, approximations of the loss with a single sample perform better than approximations with 25 samples. This coupled with the authors' observation that using with alpha-stable distributions with $\alpha < 0.5$ is "too unstable" to train underlines the fact that heavy-tailed distributions are not a practical tool for approximating distributions.
2. Extension for extension's sake: The work to extend to alpha-stable noises is theoretically interesting but appears to be an extension purely done for extension's sake. Moreover, it follows prior work that has already extended SDE diffusion models to heavy-tailed noises. The authors can strengthen their work by showing why alpha-stable noises are an interesting extension through better experiments.  The authors could define better what it means to be heavy-tailed for bounded distributions like images. Are there real distributions (in the sense of natural signals) where heavy-tailed phenomena have been observed and could the authors show DLPMs are better suited there? For example in section 4.1 why isn't DDPM included in the comparisons? In section 4.2, the FIDs reported for DDPM appear to be a little too high. The FID on CIFAR10 for EDM, a simplified DDPM, (Karras et al 2022) is 1.97, so MNIST should not be harder. In my view, the current numbers, especially in Appendix G.4, do not exhibit a strong enough advantage for DLPMs.

**Questions:**

Proposition 9: If the network is initialized close to 0 and $\hat{\epsilon}^\theta \approx 0$, the training loss derived in proposition 9 becomes an expectation of $A_t$ an alpha-stable random variable with $\alpha < 1$. Could the authors explain why this loss is not also infinite like in the LIM setting?

---

> ### Author Response · Authors · 2024-11-22
> **Rebuttal**
>
> We thank the reviewer for the thoughtful feedback and for highlighting important points for discussion. We address them below.
> * *The training loss is unstable, poorer approximations perform better: [...] the denoising network still has to learn the conditional expectation of a heavy-tailed distribution contrary to what is said in the paper.*
>
> The reviewer makes an interesting remark which indicates that we have not been clear enough in our original submission maybe. To address this point, let us inspect the loss function again:
> $$
> \mathcal{L}^{\text{simple}}(\theta) = \sum_{t=1}^{T}\mathbb{E} \left[ \mathbb{E} \left( \| \epsilon_t^{\theta}(Y_{t}) - \epsilon_{t}(Y_t, Y_0) \|^{2} \ \big| \ A_{1:t} \right)^{1/2}\right].
> $$
> From this formulation, we aim to learn $\epsilon_t(Y_t,Y_0)$ conditionally to $A_{1:t}$. Contrary to what the reviewer said, the conditional distribution of $(Y_t,Y_0)$ given $A_{1:t}$ has the same tail as the data distribution. Another perspective on our approach is to see it as a reparametrization trick, as used in variational inference, to obtain more stable and efficient training. Therefore, our training loss is stable, which is supported by our numerical experiments.
>
> If we did not address reviewer's comment sufficiently on the loss, we would be happy to provide more details on these concerns.
>
> * *approximations of the loss with a single sample perform better than approximations with 25 samples*
>
> Thank you for your this interesting comment. We suspect that, alike what happens with gradient descent vs stochastic gradient descent, the greater stochasticity is actually helping the network in some way (flatter minima, better robustness, better generalization etc.). You can think of the cases where SGD with smaller batch-size performs better than a large batch-size, even though the stochastic gradient with the small batch-size is a "poorer" approximation of the full gradient.
>
> Furthermore, additional work is needed to determine the optimal batch sizes and other parameters, which is outside the scope of this initial exploration. We would also like to mention that our paper has a strong exploratory aspect, investigating available methods for heavy-tailed diffusion. Regarding the stability of the loss and the median-of-means method, we observed a positive trend of improvements as $M$ increases, which we believe, justified its inclusion in the paper.
>
> If the reviewer finds it relevant, we can add this discussion in the revised version of the paper.
>
> * *heavy-tailed distributions are not a practical tool for approximating distributions*
>
> Regarding the instability for $\alpha \leq 1.5$, it is indeed possible to train models under such conditions, but this requires additional techniques such as gradient clipping, noise clipping, and extended training times. These adaptations align with the rationale that heavy-tailed noises explore a larger portion of the space as compared to Gaussian noises, particularly in high-dimensional settings, where Gaussian noise concentrates in thin shells around datapoints. This naturally impacts training.
>
> Our goal was to present a simple and principled framework that could be used effectively "out of the box", hence our simplifying remark.
>
> It is worth noting that, for satisfiable performance, LIM requires gradient and noise clipping introducing additional  clipping hyper-parameters. These parameters have to be tuned for every $\alpha$ and for each dataset, whereas this is not required by our method. In this sense, our method, contrary to LIM, is actually the first to make heavy-tails practical, enabling straightforward manipulation at least when $\alpha \geq 1.5$.
>
> * *In section 4.1 why isn't DDPM included in the comparison?*
>
> We added DDPM to Table 1 and Table 2. For Table 2, we increased the number of test samples to 25000 to improve statistical significance.
>
> **Table 1: $\text{MSLE}_{\xi = 0.95} \downarrow$, averaged over 20 runs**
>
> | Method     | α=1.5  | α=1.6 | α=1.7 | α=1.8 | α=1.9 | α=2.0 |
> |---|-|---|-|---|--|--|
> | DLPM       | 0.160 ± 0.128     | 0.081 ± 0.078     | 0.071 ± 0.028 | 0.099 ± 0.044 | 0.132 ± 0.101 | 0.798 ± 0.601    |
> | DDPM       | -   | -    | -    | -       | -     | 0.528 ± 0.400 |
>
> **Table 2: $F_1 \uparrow$ score, averaged over 30 runs**
>
> | Method      | α=1.5      | α=1.6    | α=1.7    | α=1.8     | α=1.9 | α=2.0 |
> |----|------|-----|-------|------|-----|-----|
> | DLPM        | 0.933 ± 0.018     | 0.923 ± 0.005     | 0.933 ± 0.028 | 0.923 ± 0.024 | 0.907 ± 0.034 | 0.862 ± 0.028    |
> | DDPM        | -   | -    | -    | -  | -  | 0.867 ± 0.029 |

---

> ### Author Response · Authors · 2024-11-22
> **.**
>
> * *Extension for extension's sake. [...] It follows prior work that has already extended SDE diffusion models to heavy-tailed noises.*
>
> Unlike prior work, our framework offers better flexibility, and a principled loss that upper bounds the log-likelihood (potentially enabling further theoretical work). We believe that the reviewer acknowledged these  aspects in his review.
>
> In addition, as emphasised in our previous response, our methodology provides a simpler setup with fewer hyperparameters, as LIM requires clipping to achieve the best performance (which is what is reported in the paper). This raises the question of the relevance of their framework for heavy-tails. We have re-run experiments for LIM without clipping on both MNIST and CIFAR10\_LT, with the same parameters. Here are the results:
>
> **Table 3: FID $\uparrow$ for our image datasets**
>
> | MNIST   | α=1.5  | α=1.7  | α=1.8  | α=1.9   | α=2.0 |
> |--|--|--|--|-|-|
> | LIM (no clip)       | 14.37  | 11.54  | 11.18  | 13.75   | -     |
> | LIM-ODE (no clip)   | 49.63  | 78.59  | 92.93  | 109.48  | -     |
>
> | CIFAR10_LT | α=1.5  | α=1.7  | α=1.8  | α=1.9   |  α=2.0   |
> |--|--|--|--|---|--|
> | LIM (no clip)       | 75.38  | 35.15  | 31.14  | 21.68   | -     |
> | LIM-ODE (no clip)   | 42.07  | 91.64  | 105.95 | 407.79  | -     |
>
> We believe this constitutes a much stronger case for the DLPM framework, and we will include this distinction in the revised paper. In conclusion, all the differences between LIM and DLPM that we exhibit makes, in our opinion, our contribution both novel and valuable, from both a theoretical and practical perspectives.
>
> * *Various concerns about heavy-tails*
>
> We want to address the reviewer's concerns about the use of heavy-tails. In the paper, we have considered that the literature has already proven the benefits of heavy-tails, as further described in [1, 2, 3], and we take it as a given.
>
> In the context of images, long-tailed data is essentially a special type of multi-class imbalanced data with a sufficiently large number of tail (minority) classes. Moreover, the combined importance of these tail classes is very significant, although each tail class by itself only has a small number of samples. Thus one can think of the data as following a sort of multi-dimensional power law. Moreover, image datasets are high-dimensional spaces, where datapoints localize in very small regions; thus, considering boundedness in this geometry does not prevent heavy-tailed behavior to emerge, especially in the case of long-tails, and in a finite data regime.
>
> In short, the long-tailed property is analogous to the presence of heavy-tails in the data. This is why DLPM and LIM are inherently more robust to mode collapse in imbalanced data.
>
> Long- tailed distributed data is a relatively common phenomenon in real-world scenarios, e.g. high-speed train fault diagnosis, escalator safety monitoring [1]. Heavy-tails are also naturally occurring in weather data [4], and in our introduction we cite financial data [2] (stock returns and factors in factor models, like BARRA [3], are heavy-tailed). Our framework could be especially relevant in these contexts. We will provide more details in the main body of the paper, if the reviewer believes it is relevant.
>
> We leave experiments on other heavy-tailed datasets to further work.
>
> [1] Zhang et al., 2024, "A Systematic Review on Long-Tailed Learning"
>
> [2] Borak et al., 2005, "Statistical Tools for Finance and Insurance"
>
> [3] MSCI, 2017, https://www.msci.com/eqb/methodology/meth_docs/MSCI_Barra_Factor_Indexes_Methodology_June2017.pdf
>
> [4] Pandey et al., 2024, "Heavy-Tailed Diffusion Models"
>
> * *In section 4.2, the FIDs reported for DDPM appear to be a little too high*
>
> EDM employs a different network architecture, noise schedules, training distribution etc., all optimized and tailored for Gaussian diffusion. Our experiments aim to isolate the effect of the noise distribution on generation while keeping other aspects constant. While DLPMs may not yet achieve SOTA performance, our work provides a foundation for future improvements.

---

> ### Author Response · Authors · 2024-11-22
> **..**
>
> * *Proposition 9: [...] why this loss is not also infinite like in the LIM setting?*
>
> This is a good remark, and fortunately there is a square-root applied to $A_t$. Let us rewrite the loss in Proposition 9:
>     \begin{equation}
>          \mathcal{L}^{\text{SimpleLess}}(\theta) = \sum_{t=1}^{T} \mathbb{E} \left[\mathbb{E} \left( \| \epsilon_t^{\theta}(Z_{t}) - \bar A_t^{1/2} \bar G_t \|^{2} \ | \bar A_t \right)^{1/2}\right].
>     \end{equation}
>     Remark that $\mathbb{E} (\|\bar A_t^{1/2} \bar G_t \|^{2} \ | \bar A_t)^{1/2} = \bar A_t \mathbb{E} (\|\bar G_t \|^{2})^{1/2} = \bar A_t \sqrt{d}$, since $\bar G_t \sim \mathcal{N}(0, I_d)$.
>     Thus, setting $\epsilon_t^{\theta} = 0$, we obtain
>     \begin{equation}
>     \mathcal{L}^{\text{SimpleLess}}(\theta) = \sum_{t=1}^{T} \mathbb{E} \left[\mathbb{E} \left( \| \bar A_t^{1/2} \bar G_t \|^{2} \ | \bar A_t \right)^{1/2}\right] =
>     \sum_{t=1}^T\mathbb{E}[
>     \bar A_t^{1/2}] < \infty.
>     \end{equation}
>     Indeed, $\bar A_t \sim \mathcal{S}_{\alpha/2,1}(0, c_A )$, so all of its moments of order $< \alpha / 2$ are defined, and we always choose $1 < \alpha \leq 2$ for DLPM and use a square root in the loss (7) ($r=1/2$) as we mention in section 3.2.3 line 309.
>
> In short, the inclusion of the square root in the loss (setting $r=1/2$ and working with $1 < \alpha \leq 2$) ensures that the loss is always finite. We appreciate the opportunity to clarify this point and if the reviewer finds it relevant, we will explicitly address this in the paper when introducing DLPM, to ensure there is no ambiguity.
>
>
> We appreciated the opportunity to address these points and thank the reviewer again for their careful review.

---

> > ### Author Response · Authors · 2024-11-25
> > **Reminder: Rebuttal Discussion Period Ending Soon**
> >
> > Dear Reviewer,
> >
> > We hope this message finds you well. We are writing to kindly remind you that the discussion period for the review process will soon come to an end. As the authors, we wanted to ensure that our rebuttal has been received and addressed.
> >
> > In particular, we would greatly appreciate your feedback or acknowledgment on the following key points raised in our rebuttal:
> >
> > * **Clarity on Loss Stability**: We clarified how our loss remains finite due to its square root formulation, addressing remarks about potential instability when $\alpha < 2$.
> >
> > * **Experimental Context for Heavy-Tailed Noise**: We elaborated on the practical implications of heavy-tailed distributions and their relevance for imbalanced and high-dimensional data.
> >
> > * **Generative Performance of DLPM (vs. LIM, DDPM)**: We provided additional clarification on LIM’s reliance on clipping and shared results from new experiments without clipping, which highlight the robustness and suitability of DLPM for heavy-tailed settings. We included DDPM in our updated experiments, and additional results on the 2d datasets for $\alpha=1.5, 1.6$. We welcome any further comments on these additional comparisons.
> >
> > If there are any remaining questions or points requiring clarification, we would be happy to provide additional responses or discuss further. Your feedback and engagement are invaluable to improving the quality of our work. Thank you very much for your time and contributions to the review process.
> >
> > Best regards,
> >
> > The authors

---

> > > ### Comment · Reviewer_Eq2q · 2024-11-27
> > >
> > > I thank the authors for their detailed rebuttal. I am grateful for their clarifications on the square root factor which I had missed and the fact that they are actually working with $1 < \alpha \leq 2$. The added comparison with LIM is welcome. I have increased my score.
> > >
> > > The following are minor, biased remarks: My concerns about the FIDs were referring to DDPM, the baseline you were comparing too, their result on MNIST for DDPM should be better. To use the heavy-tail framework for signals that are truly unbounded like some that the authors mention makes sense. To use it for images is odd in my view. I may be wrong but the long-tail paper appears like it is simply redefining terminology and stealing the term "long-tailed". I do not believe one could convincingly argue that imbalanced classes of [0,1] pixels are best modeled with distributions that have no second moment. I would focus on the datasets where it makes more sense.

---

> ### Author Response · Authors · 2024-11-27
>
> Dear reviewer,
>
> Thank you for your thoughtful comments and for taking the time to review our rebuttal in detail. We are grateful for your clarifications and are encouraged by your increased score and acknowledgment of the additional comparisons and explanations we provided.
>
> We acknowledge that DDPM’s performance can be improved, as seen in recent works, such as EDM. Even with the simple framework of DDPM, this requires some tailored modifications like changes to training regimes that goes beyond the scope of this study.
>
> Once again, thank you for your valuable feedback, which has helped us strengthen the presentation and clarity of our work.
>
> Best regards,
>
> The authors

---

### Meta-Review · Area_Chair_rpeE · 2024-12-12

**Metareview:**

The paper introduces a new diffusion framework by injecting noise following the alpha-stable distribution, which better suits heavy-tail distribution. Moreover, its implementation is simple with a small modification to DDPM, therefore offering a suite of algorithmic choices for the growing literature of diffusion models.

**Additional Comments On Reviewer Discussion:**

The initial score of this paper is somewhat mixed, but after healthy discussion during the rebuttal, all reviewers have converged to acceptance.

---

### Decision · Program_Chairs · 2025-01-22

Accept (Poster)